# Multi-year particle fluxes in Kongsfjorden, Svalbard

Alessandra D′Angelo [1,2], Federico Giglio [1], Stefano Miserocchi [1], Anna Sanchez-Vidal [3], Stefano Aliani [1], Tommaso Tesi [1], Angelo Viola [4], Mauro Mazzola [4], and Leonardo Langone *[1]

[1]CNR-ISMAR - Consiglio Nazionale delle Ricerche - Istituto di Scienze Marine, Italy
[2]Università degli Studi di Siena, Siena, Italy
[3]Universitat de Barcelona, Barcelona, Spain
[4]CNR-ISAC - Consiglio Nazionale delle Ricerche - Istituto di Scienze dell'Atmosfera e del Clima, Italy

**Correspondence:** *Leonardo Langone (leonardo.langone@ismar.cnr.it)

**Abstract.** High latitude regions are warming faster than other areas due to reduction of snow cover, sea ice loss, changes in atmospheric and ocean circulation. The combination of these processes, collectively known as polar amplification, provides an extraordinary opportunity to document the ongoing thermal destabilisation of the terrestrial cryosphere and the release of land-derived material into the aquatic environment. This study presents a six-year time-series (2010-2016) of physical parameters and particles fluxes collected by an oceanographic mooring in Kongsfjorden (Spitsbergen, Svalbard). In recent decades, Kongsfjorden has been experiencing rapid loss of sea ice coverage and retreat of local glaciers as a result of the progressive increase of ocean and air temperatures. The overarching goal of this study was to continuous monitoring the inner fjord particle sinking and to understand to what extent the temporal evolution of particulate fluxes were linked to the progressive changes in both Atlantic and freshwater input. Our data show high peaks of settling particles during warm seasons, in terms of both organic and inorganic matter. The different sources of suspended particles were described as a mixing of glacier carbonate, glacier-silicoclastic and autochthonous marine input. The glacier releasing sediments into the fjord resulted to be the predominant source, while the sediment input by rivers was reduced at the mooring site. Our time-series showed that the seasonal sunlight exerted first-order control on the particulate fluxes in the inner fjord. The marine fraction peaked when the solar radiation was maxima in May-June while the land-derived fluxes exhibited a 1-2 months lag consistent with the maximum air temperature and glacier melting. The inter-annual time-weighted total mass fluxes varied two-order of magnitudes over time, with relatively higher values in 2011, 2013 and 2015. Our results suggest that the land-derived input will remarkably increase over time in a warming scenario. Further studies are therefore needed to understand the future response of the Kongsfjorden ecosystem alterations in respect to the enhanced release of glacier-derived material.

## 1   Introduction

There is ample evidence collected over the last decades that the atmosphere and ocean have warmed, the amounts of snow and ice have diminished and sea level has risen (IPCC, 2014). Global climate change is amplified in the Arctic by several positive feedbacks, including ice and snow melting that decreases surface albedo and atmospheric stability that traps temperature anomalies near the surface layers (Overpeck et al., 1997).

The physical drivers for the changes attributed to anthropogenic climate change include the increased penetration of warm Atlantic and Pacific water into the Arctic Ocean, increased seawater temperature, reduced cover of sea ice and increased submarine irradiance (Wassmann et al., 2011). As a result, the temperature in high latitudes is increasing at a rate of two to three times than the global average temperature (ACIA, 2004). Arctic fjord systems will likely be particularly vulnerable to the human-induced climate change.

To better understand how the thermal destabilization of Arctic fjords will occur over time, it is important to establish the current knowledge for these sites. This requires the acquisition of time series for climate-sensitive parameters (Svendsen et al., 2002). Our research is part of the ARCA project (ARctic present Climate change and pAst extreme events), which aimed to develop a conceptual model on the mechanism(s) behind the release of large volumes of cold and fresh water from melting of ice caps, investigating this complex system from both paleoclimatic and modern air-sea-ice interaction process point of view.

Kongsfjorden is largely influenced by the polythermal tidewater glaciers Blomstrandbreen and Conwaybreen, as well as Kongsbreen, Kronebreen and Kongsvegen (Liestøl, 1988; Dowdeswell and Forsberg, 1992; Hagen , 1993; Svendsen et al., 2002; Howe et al., 2003; Blaszczyk et al., 2009; MacLachlan et al., 2010; Sundfjord et al., 2017). The freshwater outflow from tidewater glaciers play a double role in the biogeochemistry of the fjord, by favouring the stratification of water masses in the coastal zone during the summer melting season (Harms et al, 2007; Rajagopalan, 2012; Svendsen et al., 2002; Trusel et al., 2010), and the settling of the suspended particulate matter to the bottom, throughout flocculation (Meslard et al., 2018).

Six-years continuous data (2010-2016) have been collected using an automatic sediment trap moored in the inner Kongsfjorden. The mooring, also equipped with current meters, salinity and temperature sensors, is located between the glaciers termini and the sill receiving the influence by melt water from the glacier as well as the Atlantic Water (AW) intrusion through the southern fjord (Svendsen et al., 2002; Cottier et al., 2005).

In order to describe the downward particle fluxes of biogenic and glacier-derived material, geochemical data of sinking particles were combined with the time-series of physical environmental data measured in the inner fjord. Our overarching goal was to observe the trend of particulate fluxes over time, and to constrain the nature of sinking particles to test to what extent the composition of the trapped material is affected by climate-sensitive aspects (e.g. precipitation, glacier retreat, air and water temperature, water column mixing, etc.). In particular, by combining physical and biological parameters we tested whether the magnitude and composition of particle fluxes in the inner fjord are experiencing changes caused by the global change, and which could be the long-term effects on the biogeochemical cycles.

## 2  Study area

Svalbard is an Arctic archipelago located between 76-81°N and 10-34°E and surrounded by the Arctic Ocean in the north, the Greenland Sea to the west and the Barents Sea to the east and south. Spitsbergen is the largest island of the archipelago (Fig. 1a). Kongsfjorden is a glacially-eroded fjord, elongated in SE-NW direction, located along the west coast of Spitsbergen. It is 27-km long, varying in width from 4 km at its head to 10 km at its mouth (Svendsen et al., 2002). The inner part of Kongsfjorden is surrounded by a glacier-dominated coast with five tidewater glaciers: Blomstrandbreen and Conwaybreen,

Kongsbreen (North and South), Kronebreen and Kongsvegen (Liestøl, 1988; Howe et al., 2003; Dowdeswell and Forsberg, 1992; Svendsen et al., 2002; Blaszczyk et al., 2009; MacLachlan et al., 2010) (Fig. 1b). The Fram Strait (Fig. 1a), positioned between Northeast Greenland and the Svalbard archipelago, is the sole conduit conveying warm anomalies from the northern Atlantic to the Arctic Ocean (Beszczynska-Möller et al., 2012). In this deep strait, the upper part of the Atlantic Water (AW) becomes less saline due to the melting sea ice and the mixing with fresh surface water of Arctic origin. This allows the AW to preserve its warm core, losing less heat to the atmosphere (Beszczynska-Möller et al., 2012). The Atlantic warm core (5°C and salinity up to 35), in the form of the West Spitsbergen Current (WSC), intrudes into Kongsfjorden, passes the threshold moraine in front of glaciers close to Lovenyane, and reaches the inner part of the fjord where it comes into contact with the glacier front (Kongsvegen, Kronebreen and Kongsbreen termini) (Fig. 1b). This process modifies the WSC characteristics, producing the Transformed Atlantic Water (TAW, 1-3°C) (Svendsen et al., 2002; Cottier et al., 2005; Aliani et al., 2016). Recent changes in the relative inflow of Atlantic and Arctic waters have resulted in significant variability of environmental conditions along the western Spitsbergen shelf (Majewski et al., 2009). In recent years (1997-2010), the AW displayed variability in temperature and transport at the entrance to the Arctic Ocean, exhibiting temperature anomalies around +2°C (Beszczynska-Möller et al., 2012). This warming has enhanced melt-water fluxes from Svalbard glaciers, influencing fjord hydrology, sea ice conditions and local biota. Physical, chemical and biological processes are influenced or constrained by the local quantities and geochemical qualities of freshwater. These include stratification and vertical mixing, ocean heat flux, nutrient supply, primary production, ocean acidification, and biogeochemical cycling (Carmack et al., 2016).

## 3 Materials and methods

### 3.1 Time series data collection

A six-year (September 2010 to May 2016) data set from the mooring line Dirigibile Italia (MDI) deployed in Kongsfjorden (Svalbard) is presented. MDI was installed in the inner fjord at ~100 m depth, at the distance of 1.7 km from the southern coast, between the Kronebreen front (7 km from the southern coast) and the sill (Fig. 1). The best site to deploy the long-term data collection equipment was selected after a bathymetric and high-resolution seismic survey at GPS position 78°55'N-12°15'E. The site is a compromise between properties of the water passing across the strait, the bottom depth and the modern sediment accumulation rate. The permanent mooring MDI was first deployed in September 2010, and then serviced at annual frequency. It was equipped with a time-series Technicap sediment trap (12 receiving cups, model PPS4/3, 0.05 m$^2$ collection area) at ~20 m above the sea bottom. To prevent organic degradation during deployment, trap sample cups were filled with filtered seawater containing a 5% formaldehyde buffered solution. The interval of rotation of the sediment trap was variable between 8 and 90 days. The shorter sampling periods correspond to the summer season in which a greater variability in biological and physical/chemical conditions was expected, while the rotation intervals were longer in winter. All programmed samples were recovered, except between 3 August and 11 September 2014 due to a rotation failure. The mooring had slightly different configuration in different years (Fig. A1). The sediment trap was typically coupled with a temperature and conductivity recorder (SBE16 SeaCAT), and a single-point current meter (TRDI-DVS). From mid-2015, the current meter was substituted by an

upward ADCP (TRDI-ADCP Sentinel V100, 300 kHz) in order to obtain water current velocities over a range of depths that encompasses the entire water column and, hopefully, also the passage of icebergs or pack ice. To monitor the mid-water characteristics, a single-point current meter with a temperature sensor (Nortek Aquadopp) was tethered at ∼35 m depth. Additional high-precision thermometers (SBE56) were assembled at nominal depths of 54 m and 62 m starting from Sept. 2012 and May 2013, respectively (Fig. A1).

For safety purpose against the passage of icebergs and sea ice, the uppermost buoy was kept submerged, thus no information is available for surface water characteristics and dynamics. CTD surveys were performed each summer during the mooring servicing by using a SBE19 probe, providing additional information on the hydrological features in the inner fjord. The accuracy of the individual current speed and direction measurements is $\pm$ 1 cm s$^{-1}$ and $\pm 5°$, respectively. The accuracy of SBE16 SeaCAT and SBE56 sensors was checked against CTD casts before and after deployments. Values of solar radiation, wind speed and direction were obtained from Amundsen-Nobile Climate Change Tower (CCT) data set (Mazzola et al., 2016), while precipitation from the eKlima archive from the Norwegian Meteorological Institute (http://sharki.oslo.dnmi.no). The water temperature and salinity, the current speed and direction, together with the meteorological and radiation parameters, were measured or averaged at 30-minute intervals. Hydrographic and current meter data sets are substantially complete with only a few data gaps of the mid-water current meter (Aug. 2011, May-Aug. 2012, and Apr-Jun. 2015; Fig. A1), following battery exhaustion.

## 3.2 Trap sample treatment and analytical methods

Samples recovered from the sediment trap were stored in the dark at 4°C until they were processed at National Research Council - Institute of Marine Sciences (CNR-ISMAR) in Bologna, Italy, following the method of Chiarini et al. (2014). The organisms classified as swimmers observed in sediment trap samples were removed to avoid an overestimation of the organic content of particle flux and to reduce the bias in sediment flux analysis (Karl and Knauer, 1989). Mineral grains were also removed and considered as ice rafted detritus (IRD), deposited from iceberg melting. The amount of IRD was calculated as flux:

$$IRD flux = \frac{n_{grains}}{(m^2 \times day)} \tag{1}$$

Samples were split into sub-samples using a high precision Perimatic Premier Pump dispenser coupled with a robotic XY module to automate the splitting. At least two subsamples (100ml each) for total mass flux (TMF) determination were filtered through pre-weighed 0.45 $\mu$m filter with a mixed cellulose esters membrane, rinsed with distilled water and dried at 50°C for 24 h, then weighed. The total weight of the trapped sediment was converted to flux according to each sample duration and to the trap collection area:

$$TMF = \frac{g}{(m^2 \times day)} \tag{2}$$

The remaining aliquots were centrifuged for 10 minutes at 3000 rpm, the excess water was then removed and the tube rinsed with demineralised water to remove any remaining salt or formalin from the sediment. Samples were then centrifuged again

at 3000 rpm for 10 minutes. Finally, excess liquid was removed and the sample freeze dried. Samples were gently ground to obtain a homogeneous powder (Chiarini et al., 2014) for further chemical analysis. We determined the contents of total carbon (TC), organic carbon (OC), total nitrogen (TN) and the stable isotope compositions using a Finnigan Delta Plus XP mass spectrometer directly coupled to a Thermofisher Scientific Flash 2000 IRMS Element Analyzer via a Conflo III interface for continuous flow measurements (Kristensen and Andersen, 1987; Verardo et al., 1990; Tesi et al., 2007). OC contents and stable isotopes were measured on freeze dried samples after $CaCO_3$ removal with an acid treatment (HCl, 1.5M). Organic matter (OM) content was estimated as twice the OC content. Carbonate content was calculated assuming all inorganic carbon (%TC - %OC) was in the form of $CaCO_3$, and using the molecular mass ratio 100/12. The average standard deviation of each measurement, determined by replicate analyses of the same sample, was $\pm$ 0.07% for OC and $\pm$ 0.009% for TN. Isotopic composition of organic carbon is presented in the conventional $\delta$ notation and reported as parts per thousand (‰):

$$\delta^{13}C = \left[ \frac{\left(^{13}C/^{12}C\right)_{sample}}{\left(^{13}C/^{12}C\right)_{PDB}} - 1 \right] * 10^3 \tag{3}$$

Uncertainties were lower than $\pm$ 0.05‰, as determined from routine replicate measurements of the reference sample IAEA-CH7 (polyethylene, -32.15‰ vs VPDB). Errors for replicate analyses of the standards were $\pm$ 0.2‰. Biogenic silica ($SiO_2$ $0.4H_2O$) content was analysed using a two-step 2.5 h extraction of 20 mg of freeze-dried sample with a 0.5M $Na_2CO_3$ solution at 85°C followed by the measurement of dissolved Si and Al contents in both leachates with a Perkin-Elmer Optima 3200RL Inductive Coupled Plasma Optical Emission Spectrometer (ICP-OES) at University of Barcelona. The Si content of the first leachate was corrected by the Si/Al ratio of the second one in order to correct for the excess Si dissolved from aluminosilicates (Kamatani and Oku, 2000; Fabres et al., 2002; Ragueneau et al., 2005). Biogenic silica was transformed to opal by multiplying by 2.4 (Mortlock and Froelich, 1989). As the total composition of a sample is the sum of biogenic and lithogenic components, the percentage of lithogenic material was obtained assuming the following relationship:

% lithogenic = 100 - (% OM + % $CaCO_3$ + % opal).

### 3.3 Principal component analysis (PCA)

A Multivariate analysis (PCA) was carried out using the vegan (v. 2.4-2) package in R. The PCA was applied on the transformed dataset standardized to a mean of 0 and standard deviation of 1 to visualise intern-specific differences and correlations. Environmental variables were tested for skewness (due to strong seasonal gradients), and heavily left or right skewed data were log transformed log (x+1) to stabilise variance (all data excluding: solar radiation, salinity, air and water temperature). The variables used for the principal component analysis were: total mass flux (Mass Flux), organic carbon content (OC), $\delta^{13}C$ (C13), inorganic carbon content (IC), opal content (Opal), atmospheric temperature (AirT), solar radiation (Rad), rain precipitation (Precip), wind speed and direction (WindSpeed, Wind dir), bottom water salinity (Sal), bottom Water Temperature (WatT). The $\delta^{13}C$ data was transformed as log(x+26) to ensure positive values. A Spearman's rank correlation matrix was carried out on the datasets to see which could be combined. Missing values for air temperature (n=8) and precipitation (n=6) were calculated based on monthly averages from the rest of the time series. Environmental data were normalised to a mean of 0 and standard deviation of 1 before analysis. Euclidean distances were used to determine spatial ranges within this data.

## 4 Results

The current manuscript focuses on the time-series of downward particle fluxes in the inner Kongsfjorden. The environmental variables discussed in the text (i.e. meteorological and oceanographic variables) are included in the discussion although a complete analysis of these parameters will be showed elsewhere (Aliani et al., in prep).

### 4.1 Solar radiation

The solar radiation exhibited a clear seasonal trend, with high values in spring and summer, and near-zero values from November to January, i.e. during the polar night, as expected due to the location of the fjord (Fig. 2a). Monthly means show higher values from May to July, ranging from about 150 W m$^{-2}$ to about 270 W m$^{-2}$, depending on the year. On an annual basis, the year with more insolation (2015) exceeds that with lowest insolation (2013) by about 13% (84 to 74 W m$^{-2}$ are the yearly means), being those values associable to the cloudiness conditions and influencing snow and ice melt. Regarding the seasons, summers presented extreme values equal to 150 W m$^{-2}$ (2013) and 192 W m$^{-2}$ (2011), spring values ranging from 117 W m$^{-2}$ (2016) to 135 W m$^{-2}$ (2012), and autumn from 9 W m$^{-2}$ (2014) to 17 W m$^{-2}$ (2010).

### 4.2 Rain

The rain (precipitation with air temperature > 0°C) can help us to make inferences on the importance of surface runoff (Fig. 2a), since hydrological measurements are scarce or absent on the numerous small waterways. Daily rainfall events during the studied period were generally characterised by low intensity (on avg., 3.1 mm d$^{-1}$ excluding dry days). The maximum event was recorded in late December 2015 (26.5 mm) during an exceptionally warm winter week. A secondary peak occurred in September 2011 (25.4 mm). Although rainfalls were mostly low, events of long duration punctuated the analysed time series (e.g., 25 days in June 2013). Rainfalls showed a typical seasonal pattern with more frequent events during warmer seasons, especially in early September (Fig. 2a). Inter-annual variability was large during all the time interval studied , where summer 2013 was the wettest (cumulative values 2-3 times higher than 2011 and 2015, and 2012 and 2014, respectively). Finally, in contrast with overall pattern, during winter 2015, some events of rainfalls occurred in concomitance with above zero air temperature events.

### 4.3 Wind

The wind speed showed a very wide variability, with the highest value reaching 26 m s$^{-1}$ in February 2015 (Fig. 2b). The general mean was $4.2 \pm 2.9$ m s$^{-1}$. Generally speaking, during dark periods wind systematically blew over 4 m s$^{-1}$, while in summer less windy conditions prevailed. The most frequently observed wind direction was from south-east (120°, Fig. 3a), along the fjord axis and from the tide glacier fronts of Konsvegen and Kronebreen (see Fig. 1). Its intensity changed very little during the years and only in 2011 and 2015 there were higher frequencies of values >15 m s$^{-1}$. The second highest direction frequency was from south-west (225°). Its highest magnitude was observed in 2012, whereas higher frequencies of this direction occurred in 2010, 2013 and 2014. Finally, a less frequent wind direction was recorded from the north-west (320°)

from the outer fjord. It rarely showed values > 15 m s$^{-1}$. The wind conditions in Kongsfjorden were greatly governed by orographic steering of the large-scale wind fields and katabatic winds with cold air from the inland glaciers to the warmer fjords.

### 4.4  Air temperature

Air temperature modules the surface ice melting, and hence also the englacial and subglacial water drainage network. Similar to solar radiation, also air temperature showed clear seasonal oscillations (Fig. 2b), with higher values in summer (up to 12°C), and lower in winter (down to a minimum of -24.4°C). Nevertheless, there were important differences. First, minima and maxima of air temperature were delayed with respect to solar radiation about 2 months. Second, during winter several intrusions of mild air masses occurred, except in winter 2011, the coldest season of our time-series.

### 4.5  Oceanography

The water temperature measured at 35 m water depth (intermediate waters) exhibited the lowest value in February 2011 (-1.9°C) and the highest in August 2015 (6.9°C), showing a very wide range of values (Fig. 2c). Values close to the freezing point were measured also in February 2015. Relative warm peaks were usually recorded in late summer (August to September) each year, while temperature minima occurred in a longer time period (Jan. 2011, 2012 and 2014; Feb. in 2012 and 2015; March in 2013

and 2016). In the available time series, the minima of temperature showed an oscillatory pattern with values close to -1.9°C in 2011, 2013 and 2015, and minimum values 1-2°C higher in 2012, 2014 and 2016. Indeed, winters 2012 and 2014 showed a further peculiar character with a twin cold peak interrupted by a relatively warm time interval. Temperature maxima recorded each year did not follow the same pattern. It was around 3°C in 2010, it exceeded 4°C in 2012, but it was always higher than 6°C in remaining years. Water temperatures measured at the near-bottom (Fig. 2c) exhibited the highest value in October 2013 with

6.3°C and lowest value in February 2011 and 2015 (-1.8°C). The average values at the intermediate (2.2 ± 2.0°C) and bottom depths (2.1 ± 1.7°C) were very similar, although the near-bottom level presented slightly less dispersed values (Fig. 2c). The oscillatory pattern between the warm peaks in summer and the minima in winter were not perfectly coupled between the two levels, and often the peak value was reached earlier at the intermediate level with respect to the near bottom one, both for highest and lowest peaks. This time lag varied up to 3 months for the warm peaks of 2011 and 2015. Superimposed to the oscillating

pattern of temperatures, a general increasing trend was observed at the two levels. Based on almost 100,000 measurements for each time series, we calculated an increasing rate of 4.3°C and 1.6°C per decade at intermediate and near-bottom levels, respectively. By way of comparison, the increase of the air temperature at the Amundsen-Nobile Climate Change Tower was estimated in 3.0°C per decade over the period 2010-2017 (M. Mazzola, personal communication). Salinities recorded at the near bottom showed the highest value in December 2015 (35.28) and its lowest in July 2011 (34.10). Also salinities showed

periodic oscillations (Fig. 2d), although in this case the data range is narrower and the trend more confuse, with a not well defined seasonal signal. In contrast, the long-term trend is well apparent with an increase rate of salinity of 0.7 per decade. The current speed at mid-depth remained quite constant during the years (Fig. 2b) with a mean value of 4.4 cm s$^{-1}$. The highest values each year varied from 20 cm s$^{-1}$ in February 2014 and January 2016, to 44 cm s$^{-1}$ in March 2013, with intermediate

values of 23-27 cm s$^{-1}$ in falls 2010, 2011 and 2012 and January 2015. Due to low current speeds, current directions were highly variable from year to year. Nevertheless, current direction was mainly toward the inner part of the fjord (Fig. 3b) with the exception of 2015, when prevailed a southward direction.

## 4.6 Particle fluxes

Fluxes of total particulate matter in Kongsfjorden showed a clear seasonal trend (Fig. 2e) with high values in summer and low fluxes during rest of the year. The mean TMF was $32 \pm 57$ g m$^{-2}$d$^{-1}$. The highest TMF was recognised from 19 to 27 August 2013 with a peak value of 330 g m$^{-2}$d$^{-1}$. Actually, all samples collected in August and early September 2013 were characterised by TMF > 200 g m$^{-2}$d$^{-1}$, making summer 2013 the period with the maximum cumulative particle accumulation. After that, a secondary, more limited maximum peak in TMF was recorded in August 2015 (123 g m$^{-2}$d$^{-1}$). In contrast, peak TMFs in summers 2011 and 2012 were much lower, such as those measured in 2014, although a mechanical failure during July-August 2014 prevented to obtain a complete time series of sediment bottles.

## 4.7 Particle composition

Temporal variation of the major components of the total mass flux are shown in Fig. 2f and g. The lithogenic material was the principal component of the flux representing almost the three-quarters of the material (range 64-78%), with small seasonal and inter-annual variability. Carbonates were the second most abundant components in sediment trap samples (18.4-31.5 %, mean value $23.5 \pm 2.6$ %CaCO$_3$). Slightly higher concentrations of carbonate were measured during periods of high TMFs (summer), especially in August-September 2013. Due to the high average values coupled with scarce seasonal variability, the major part of this component seems not to be derived by shells of living carbonatic organisms. Organic carbon (OC) and opal contents never exceed 10% in abundances and showed a marked seasonal variability. Peak values of OC contents varied between 2.4% in 2016 and 5.1% in 2015) (Fig. 2g), whereas opal maxima ranged between 1.5% in 2016 and 7.3% in 2013. The timing of abundance peaks of the two biological components were similar, usually showing the highest values in April-June, always anticipating the TMF peaks (Fig. 2e and f). Actually, in 2013 and 2015 the OC peaks occurred one month delayed with respect to opal ones. During the remaining part of years, opal contents were negligible, whereas OC varied between 0.3 and 0.7%. $\delta^{13}$C (Fig. 2h), measured in the organic fraction of C, varied from -25.9‰ to -21.4‰ (mean, -23.5‰), showing a clear temporal trend, which mirrored that of OC contents. For heavier $\delta^{13}$C values, OC contents were the highest and vice versa. The temporal pattern of IRD differed from those of other components, showing neither a clear seasonal change, nor inter-annual constant fluctuations (Fig. 2g). Nevertheless, its variability displayed some peculiarities worthy of being described. IRD abundances displayed a peak value of 600 items m$^{-2}$d$^{-1}$ in August 2013, parallel to the highest TMF. However, mean IRD abundance dropped after September 2013 from 195 items m$^{-2}$d$^{-1}$ to 68 items m$^{-2}$d$^{-1}$. Furthermore, the first 3-years of the time series were characterised by a more pronounced temporal variability, punctuated by at least 4 time intervals with higher IRD contents. Although they roughly occurred in concomitance with the peaks of TMF, their duration was longer, starting earlier and decreasing after those of TMF. C/N ratio varied between 6.0 and 13.8 (mean, $9.3 \pm 1.5$) in a relative narrow range. Higher values generally mirrored those of TMF, while lower C/N were in agreement with the OC maximum concentrations

(Fig. 2h). Exceptions to this pattern were the high values of C/N ratios in Feb. 2011, March 2012, Jan. 2013, and at lesser extent, March 2014.

## 4.8 PCA

The environmental data were described through a multivariate analysis (PCA) to summarise biotic and physical data in principal components. The PCA was performed on the sediment trap samples (n = 73) together with the environmental parameters averaged according to the time intervals of sediment trap samples (Table A1, Appendix). Table 1 shows 12 PCs which accounts for 100% of the variation, with PC1 and PC2 accounting for 55.5%. The coefficients of variables for PC1 and PC2 are shown in Table 2. The major coefficients for PC1 are: organic and inorganic carbon (OC, $\delta^{13}$C and IC), Opal, total mass flux (MassFlux). The coefficients which displayed strong positive loadings are: organic matter (Opal, OC and $\delta^{13}$C). Total mass flux, together with inorganic carbon showed the opposite (Table 2). The main coefficient for PC2 is the wind speed, whereas air and water temperatures, radiation, precipitation, (weakly) salinity and wind direction are in opposite loading (Table 2).

## 5 Discussion

In the last decades, many studies focused on the identification of sediment sources in the Kongsfjorden system were carried out by using different approaches (Trusel et al., 2010; Svendsen et al., 2002; Kuliński et al., 2014; Kim et al., 2011; Lalande et al., 2016a). Some studies investigated the entire fjord, others concentrated in its inner part, at the interface between fjord waters and the glacial fronts. A few studies included seasonally-occupied stations in order to have a complete annual cycle (e.g., Lalande et al., 2016a; Wiedmann et al., 2016). However, multi-year time-series of particle fluxes in the inner fjord has not yet been obtained, in this respect, our study is novel. The marine ecosystem of Kongsfjorden experiences pronounced seasonal variability in sunlight, glacier melt, and ice cover. As a consequence, large variations in vertical particle fluxes due to primary production (export fluxes) and lateral advection of detrital particles were expected (Lalande et al., 2016a). In the next sections, size and composition of sediment trap data will be discussed in order to elucidate which are the main atmospheric, oceanographic, biological or sedimentological factors capable to modulate the seasonal and inter-annual variability of particle fluxes. In addition, we will focus on the full 6-year time-series and, using complementary physical data, try to understand what processes trigger events of exceptional high particle flux.

## 5.1 Seasonal variability of particle fluxes

Our 6-year time-series of downward particle fluxes show a marked seasonal variability, confirming findings by Lalande et al. (2016a). The typical seasonal pattern of particle fluxes can be summarised as follows:

- Spring: dominated by the export of OM with high opal and OC contents. Maximum fluxes of TMF, opal and OC were recorded in 2013 (Avg. TMF, $9 \pm 8$ g m$^{-2}$d$^{-1}$, $163 \pm 172$ mg OC m$^{-2}$d$^{-1}$, $206 \pm 436$ mg opal m$^{-2}$d$^{-1}$);

- Summer: maximum values of TMF (330 g $^{-2}$d$^{-1}$) consisting of lithogenic and carbonate sediments (Avg. TMF, 85 $\pm$ 81 g m$^{-2}$d$^{-1}$, 318 $\pm$183 mg OC m$^{-2}$d$^{-1}$, 11 $\pm$ 27 mg opal m$^{-2}$d$^{-1}$);

- Polar dark night: low organic and inorganic fluxes (Avg. TMF, 10 $\pm$ 6 g m$^{-2}$d$^{-1}$, 66 $\pm$ 31 mg OC m$^{-2}$d$^{-1}$, 3 $\pm$ 7 mg opal m$^{-2}$d$^{-1}$).

In summary, TMFs in summer were about one order of magnitude higher than the rest of the year. Opal fluxes in spring were from one to two orders of magnitude higher than in summer and polar night, respectively. OC fluxes showed a relatively narrower seasonal variability: maximum in summer for the contribution of mixed (fresh and ancient) organic matter, about one half in spring due to export from the upper layer of new-produced phytoplankton cells, and $\sim$20% of the summer fluxes during the polar night, essentially constituted by ancient OM. In order to understand to what extent the physical variables affect the

environment, we have converted a set of biotic and physical observations of possibly correlated variables into a set of values of uncorrelated variables called principal components. Based on the distribution of the variables, the first axis (PC1, 30.9% of the variance) likely describes the mixing of two different particulate sources associated to springs and summers, respectively (Fig. 4). The former was rich in freshly produced autochthonous material (high OC, opal and $\delta^{13}$C values), while the latter is primarily associated with the input of glacier-derived material (high mass fluxes and high inorganic C). In addition, while

the radiance explains the high-productive periods, the land-to-fjord exchange seems to be a function of the temperature in the fjord (high air and water temperature). Finally, the spring samples that show low loadings reflect periods characterised by low primary productivity associated with pre-algal bloom period while fall samples with negative loadings are probably related to protracted warm periods in early fall. The second axis (PC2, 24.6% of the variance) reflects the general seasonal trend and is dominated by the physical parameters. Along this axis, samples grouped in two main clusters: spring-summer and

fall-winter. The former period is characterised by positive change in precipitation, temperatures, and radiance all typical of the spring-summer periods, while fall-winter mainly described by the enhanced wind speed with, however, little influence on the mass flux and its composition. As Kongsfjorden is experiencing an ever-increasing warm scenario, our results from the PCA suggests that the land-ocean flux of glacier-derived material will greatly increase, while the biotic component might change to a minor extent reflecting primarily changes in light availability (e.g., sea-ice occurrence).

## 5.2   Nature of collected particles

Particles intercepted by our sediment trap mainly consisted of silicoclastic material (Fig. 2f). Carbonates were the second most abundant component. The source of this fraction could be either biogenic, due to the contribution of organisms with carbonate shells, or clastic, as a result of erosion of old carbonate rocks by the glaciers. The PCA provides further details about the CaCO$_3$ origin and indicates that CaCO$_3$ dominates during period of high mass flux associated with the high summer temperatures,

while fraction of inorganic carbon is relatively low during the high productive season. Furthermore, the CaCO$_3$ contents were not related to the swimmer *Limacina helicina* abundances, the most abundant swimmer with carbonate shell in our sediment trap samples (D'Angelo et al., in prep.). Altogether, this implies that most of carbonate grains collected by our sediment trap is detritic, consistent with high CaCO$_3$ concentrations observed in inner fjord sediments close to the glacier front (Bourgeois

et al 2016). Combined, silicoclastic sediment and clastic carbonates represent > 90% of material accumulating in the inner Kongfjorden (Fig. 2f). In the catchment area of Kronebreen and Kongsvegen glaciers, unconsolidated deposits of Quaternary age include the Gipshuken Fm with dolomite breccia, dolomites, sandstones and marl, and the Wordiekammen Fm (Dallmann, 2015). The latter contains limestones and dolomite, which are rich in calcium. Due to the geographical vicinity and composi-

tional affinity, the Kronebreen and Kongsvegen glaciers seem thus the most probable sources of clastic sediment for the MDI site. To investigate the main sources that contribute to supply organic particles accumulating in the studied area, the elemental and isotopic composition of organic matter were considered, by plotting $\delta^{13}$C versus OC/TN molar ratios (Fig. 5). In the Arctic realm, C/N ratio (Stein and Macdonald, 2004) and $\delta^{13}$C (Schubert and Calvert, 2001; Winkelmann and Knies, 2005) have been widely used as proxies to discriminate between marine and terrestrial provenance of organic material (Hedges and Oades,

1997). First results have indicated that the sediments in the Kongsfjorden-Krossfjord system primarily contain organic matter of marine origin ($\delta^{13}$C, -20.6‰) with respect to that of soil and coal origin (Svendsen et al., 2002; Winkelmann and Knies, 2005). Through organic chemistry analyses, and $\Delta^{14}$C values, Kim et al. (2011) have suggested that ancient OM of both coal-derived and mature glacial OM is being buried in the Kongsfjorden-Krossfjorden system. Kuliński et al. (2014) tried to apply a 3-end-member mixed model, where the first OM source was ancient from subglacial drainage. The second one was fresh ter-

restrial OM from river discharges, and the third end-member was constituted by fresh marine phytoplankton. They provided the value of $\delta^{13}$C for the first two end-members, but found a high range of $\delta^{13}$C for the marine end-member, which precluded the use of the mixing model (Kuliński et al., 2014). Bourgeois et al. (2016) showed an OM composition gradient from the inner to outer Kongfjorden reflecting a decreasing glacier contribution and a concurrent marine fraction increase. $\delta^{13}$C values in organic matter of Kongsfjorden sediments varied between -23.8 and -19.3‰, with the heaviest values measured in sediments collected

in front of glaciers. A similar spatial variability has been found also by Kumar et al. (2016), where the marine organic matter was unusually more depleted in $\delta^{13}$C ($\sim$-24‰) than the terrestrial organic matter ($\sim$-22.5‰). On the other hands, Calleja et al. (2017) indicated an input of terrigenous POC material coming from the turbid plumes of melting glacial ice during August and October characterized by $\delta^{13}$C values consistently depleted in $\delta^{13}$C (ranging from -25.6 to -27.1‰) and the highest value of -22.7‰ occurring in May at the peak of Chl a concentration, consistent with phytoplankton end-members commonly used.

These apparent discrepancies could be reconciled in view of the higher temporal variability of stable isotope composition with respect to the spatial one. The distribution pattern of C/N and $\delta^{13}$C values of our samples (Fig. 5) shows a quasi-triangular shape, suggesting three main sources of OM collected by the sediment trap of mooring MDI. Consistent with our early discussion based on the PCA results, the first end-member is marine particulate material characterised by low C/N ratios, relatively heavier $\delta^{13}$C values and high opal and OC contents (Fig. 5). The second end-member is instead associated with the material

released by the glacier and it characterised by high TMF and C/N ratios, relatively low $\delta^{13}$C values, lack of opal and rich in $CaCO_3$ (Fig. 5). The third end-member remains, though, elusive. Low opal contents do not support the hypothesis of in-situ diatom production while the relatively high OC content would suggest glacier outflows quantitatively enriched in fossil/subfossil bioavailable carbon since glaciers may be eroding ancient peatlands (Hood et al., 2009); in fact, during the warmer parts of Holocene many of Svalbard's glaciers retreated onto land and proglacial areas were covered by tundra (Lydersen et al., 2014).

Overall, the depleted $\delta^{13}$C values might suggest a land-derived origin. However, the isotopic fingerprint of the largest river

(Bayelva, Fig. 1) discharging into the fjord displays relatively heavier compositions (range -24.3/-23.5‰) (Zhu et al., 2016). In addition, although the coal-derived OM is another potential source due to the mining activity occurred over the last century, its input is mainly constrained around the region adjacent to Ny-Ålesund coast (Kim et al., 2011) and, anyhow, it would result in remarkably high C/N ratios. Finally, the contemporary input of soil OM into the fjord can be considered negligible, likely due to the limited soil formation in the cold Arctic environment, which weakens the signature of soil OM in marine environments (Kim et al., 2011). Altogether, our results might suggest the input of a second OM pool from the glacier in addition to the carbonate-rich end-member. This input, generally referred to silicoclatic-rich material, might be associated with a different glacier or rock source. However, additional analyses (e.g., biomarkers) are needed to further refine the third OC source to the inner fjord.

## 5.3 Factors governing particle fluxes of marine origin

Elevated vertical fluxes of fresh OM of marine origin, as well as high OC and opal contents (Fig. 2f and g) in spring indicate the occurrence of substantial algal blooms (Lalande et al., 2016a). Typically, Kongsfjorden spring blooms peak in May and consist of diatoms and *Phaeocystis pouchetii* (Hegseth and Tverberg, 2013). The higher proportion of opal with respect to OC (Fig. 6) in spring samples corroborates the interpretation of periods characterized by high diatom export fluxes, whereas in time intervals with lower opal/OC ratios, a mixed phytoplankton composition probably prevails. The phytoplankton growth is mainly driven by the availability of light and nutrients, although other factors can influence algal dynamics (e.g., water temperature, zooplankton grazing, water stratification, etc.). In the Arctic region, it has been established that seasonality is manly driven by the light regime and the angle of the sun, rather than temperature (Berge et al., 2015). The seasonal pattern of fresh marine OM of our time-series clearly shows that export fluxes in April-June (Fig. 2g) were closely coupled with the algal blooms, which occur when light is enough to trigger the primary productivity (Fig. 2a), suggesting that in Kongsfjorden solar radiation exerts a first-order control on the timing of phytoplankton blooms and on the following export fluxes. The nutrient availability is instead less predictable than the seasonal irradiance and in the ocean it can be a function of the water mixing, which results in the intrusion of nutrient-rich waters into the euphotic zone. To test if convective mixing takes place during winter in Kongsfjorden, we analysed a time-series section of water temperature (Fig. 7) extrapolated by measurements recorded at 4 levels (36-37 m, 54-58 m, 62-69 m, 87 m) between May 2013 and June 2016. Vertical temperatures showed large temporal differences. Overall, warmer periods occur in late summer and fall. In contrast, colder water periods characterise winter and early spring seasons. In both cases, the portion of water column investigated (30-90 m) appears stratified. The shift between the two conditions is rapid with a complete mixing of the water column. The greater intrusion of AW at mid-depth is probably responsible of the heating of the interior of the water column, while the cooling of surface waters during winters with the formation of local dense waters by low air temperatures and the high wind regime triggers the convective mixing and the sinking of the new formed waters to the bottom. However, year-to-year differences were recorded with winter AW intrusions in Jan-Feb 2012, 2014, and 2016 (Fig. 2c and 9), which prevented a complete mixing of the water column and, presumably, a less efficient upward nutrient supply. When this occurs, the relatively less abundance of nutrient results in low primary productivity in the inner fjord. If our interpretation is correct, then the inter-annual variability of spring export fluxes

in Kongsfjorden is driven by nutrient supply and, ultimately, by the different year-to-year winter intrusion of Atlantic water. The anomalous inflow events of warm Atlantic water in winter can also prevent sea ice formation in Kongsfjorden, delaying the bloom timing by 3-4 weeks in spring (Hegseth and Tverberg, 2013). Specifically, three scenarios have been described (Hegseth and Tverberg, 2013):

a ) When there is sea ice in the fjord during winter, and lots of ice outside, with a normal inflow of AW along the bottom. This leads to a May bloom;

b ) When there is open water in the fjord in combination with ice outside the fjord, with a normal inflow of AW along the bottom. This leads to an April bloom;

c ) When both fjord and adjacent shelf are open, and there is an unusual surface inflow of AW to the fjord in winter. This leads again to a May bloom.

During the investigated period, Kongsfjorden was covered by sea ice only in winter 2011, thus we expected that the algal bloom occurred in May. In the following years, the sea ice coverage was negligible (Payne, 2015). In 2012, 2016 and especially in 2014, due to the incursion of AW between January and March, the algal blooms were probably delayed following the last scenario (c). This is confirmed by the modest OM export in our sediment trap (Fig. 2g). Finally, the algal blooms in 2013 and 2015 followed the scenario (b), with export fluxes occurring earlier, in late April. In summary, the seasonal signal of export fluxes is triggered by solar light. The onset of OC export can change from year-to-year from late April to late May (Fig. 2g) depending on absence or presence of sea ice cover and on the AW intrusion during the previous winter. The winter incursion of AW into the fjord also affects the input of nutrients from bottom waters modulating the magnitude of the primary production and the corresponding inter-annual variability of export fluxes.

## 5.4 Lateral advection of clastic particles: temporal variability, sources and mechanisms of delivery

The most part of downward particle fluxes is restricted to a short period in the summer: in July-September, the mass flux accounts for 38% to 71% of the annual flux, depending on the collection year. As previously described, particles in summer are almost completely constituted by terrigenous sediments (on avg., ~99%), both carbonate and silicoclastic. In line with what shown by the PCA (Fig. 4), the total mass fluxes peak in July-August, after the solar radiation maxima (late June) (Fig. 2a), but perfectly in phase with air temperature (Fig. 2b) and partially with rain (Fig. 2a). The highest water temperature values are instead slightly delayed, occurring in August to September (Fig. 2g). In the previous paragraphs, we inferred that glaciers are the most probable source of clastic sediment for site MDI. Air temperature strongly affects glacier drainage, which in summer begins once there is sufficient surface melt to warm the glacier snowpack to the melting point, allowing water to make its way downward to the subglacial drainage network (Lydersen et al., 2014). As surface water enters this subglacial network, overpressure occurs, leading to lift-up and entrainment of basal sediments. As a result, the water coming out of the glacier tends to be sediment-laden (Hodgkins et al., 1999). On the contrary, when air temperature decreases below zero, during winter months, the processes of meltwater refreezing and water storage switch to a slow drainage system (Lydersen et al.,

2014). This produces a pronounced seasonality in freshwater (Jansson et al., 2003) and sediment discharge, and ultimately in downward particle fluxes in the inner fjord, as recorded by our sediment trap (Fig. 2f). In order to test if meltwater runoff is the most important factor driving the suspended matter supply to the fjord, we used positive degree days (PDDs) as a proxy for meltwater availability. Following the methods of Schild and Hamilton (2013), the daily average temperature after the onset of

melt is defined as the PDD value for each day. To take into account lags in the hydraulic system introduced by finite transit time of meltwater through the glacier system and potential subglacial storage, we calculated a 'lag index' using accumulated PDDs. This index is the cumulative sum of the PDDs in the 6 days prior to that day (Schild et al., 2017). The perfect synchronicity of TMF and accumulated PDD (Fig. 8) suggests that glacial meltwater runoff, the subglacial transport of meltwater, is the main process able to supply clastic sediment to Kongsfjorden, and that the seasonal variability of air temperature, specifically the

accumulated temperature above the melting point, ultimately modulates also the timing of TMF at site MDI.

It has been recently shown (Luckman et al., 2015) that ocean temperature strongly impacts the calving rates of tidewater glaciers. In Kongsfjorden, submarine melting is specifically forced by the AW intrusions. However, the delay of maximum water temperature (Fig. 2c) from the TMF peaks (Fig. 2f) would suggest that submarine melting is a minor font of sediment for site MDI, and can at best explain the slow declining trend of TMF during some falls (e.g., in 2011, 2013 and 2015; Fig. 2f).

Marine currents measured at mid-depth of mooring MDI were mostly directed toward the inner fjord (Fig. 3b), whereas winds coming from the glacier valleys (Fig. 3a) force the surface water to move out of the fjord (Aliani et al., 2016). It results that water circulation in the innermost part of Kongsfjorden is of estuarine-like type with cold freshwater on top of warm saline water (Svendsen et al., 2002; Cottier et al., 2005; Trusel et al., 2010; Aliani et al., 2016; Sundfjord et al., 2017). This type of circulation implies that the glacially-derived sediment will spread at surface through hypopycnal sediment-rich plumes

dispersed by winds over the fjord. Specifically, the primary driver of hypopycnal plume formation is a meltwater upwelling from Kronebreen (Trusel et al., 2010). As meltwater discharges from the base of the glacier, it entrains high concentrations of sediment and rapidly forms a turbulent jet (Powell, 1991). Because of relative density contrasts, the jet rises vertically and forms a buoyant and brackish surface overflow plume. At the surface, the brackish plume spreads then laterally (Trusel et al., 2010) driven by the wind.

Finally, PCA results (Fig. 4) suggest that even the surface runoff by rainfalls can increase the sediment delivery to the fjord, especially in summer 2013 (Fig. 2a), making the upper water very turbid and resulting in typical red coloured waters. Winter precipitation events in Ny-Ålesund have been increasing during recent years (on avg. 3-4% per decade for the 1961 - 2010 period (Førland et al., 2011). As a matter of facts, the concurrent air temperature increase may enhance the rain fraction against snow. The rain events recorded in Jan-Feb 2016 (Fig. 2a) testify the incursion of mild and wetter air masses into the

fjord. Therefore, in a warming scenario, an increase of particle flux to Kongsfjorden by local watersheds can be expected, even during winter.

## 5.5  Annual fluxes and possible changes

TMFs measured at the MDI station are about 20-100 times higher compared to downward fluxes measured on the Spitsbergen continental margin (Sanchez-Vidal et al., 2015; Lalande et al., 2016b), but of the same order of those measured in August 2012

in Kongsfjorden by Lalande et al. (2016a). For studying the variability of particle fluxes at multi-year scale the use of annual time-averaged TMFs is preferable. The lowest time-weighted averages were calculated for 2010, 2014 and 2016 ($4.6 \times 10^3$; $3.8 \times 10^3$; $3.1 \times 10^3$ g m$^{-2}$ y$^{-1}$, respectively), but these time-weighted TMFs could be underestimated because the time-series were not complete and missed the summer, the most abundant flux season. The inter-annual time-weighted TMFs varied by a
factor of 2 over time ($6.8 \times 10^3$ to $14.7 \times 10^3$ g m$^{-2}$ for 2012 and 2013, respectively).

    Annual total mass and main component fluxes do not show any clear temporal trend (Fig. 9). In 2013, the highest fluxes resulted from the combined effect of enhanced subglacial runoff (as pointed out by air temperature and PDD), submarine melting (water temperature), and surface runoff (rain). In contrast, IRD fluxes strongly decrease over time (Fig. 9). IRD abundance in sediment cores is commonly used in paleoceanographic reconstructions as a proxy for iceberg production and transit. How-
ever, the water temperature affecting the calving rate, and hence the iceberg production, showed an increasing temporal trend in Kongsfjorden during 2010-2016, which would imply an enhanced submarine melting. On the other hand, annual TMF and IRD fluxes in 2013-2015 seem well correlated supporting a common source from glacier melting (Fig. 10), while an additional source of IRD must be taken into account for years 2011 and 2012. At least for 2011, when the fjord froze for the last time, we tentatively suggest that the source of this extra IRD may be linked to the summer melting of "dirty" sea ice formed in the
previous winter by suspension freezing or entrainment via anchor ice (Darby et al., 2011). Alternatively, we can argue that the sediment deposits at Kronebreen and Kongsvegen glacier termini reduced the relative depth of water over time, resulting in increased glacier stability through decreased rates of iceberg calving (Kehrl et al., 2011), and hence also lower IRD fluxes along the iceberg transit pathway.

    As previously discussed, the variability of biogenic fluxes (organic matter and opal) mainly depends on the nutrient supply
and indirectly on the efficiency of convective mixing during winter as a balance between the winter AW intrusion and local dense water formation. Our 6-year time series is still too short for obtaining reliable long-term prediction, but the increasing trends in salinity (0.7 per decade) and temperature (1.6°C to 4.3°C per decade at near-bottom and mid-depth, respectively) are objectively clear, such as the year-to-year alternation of deep and shallow convective mixing. The warming of the Atlantic Water entering Kongsfjorden would most likely continue to hinder the formation of ice cover, contributing to phytoplankton as-
semblage changes, but also to more frequent conditions of shallower water mixing and lower nutrient supply, with a consequent expected reduction of magnitude of primary production and export fluxes. This scenario has already occurred in Adventfjorden during 2007-2008 (Zajączkowski et al., 2010). Although the increase of the air temperature at CCT is a straight signal over the period 2010-2017 (Fig. 2a), the accumulated PDD, as well as TMFs varied more confusedly. Nevertheless, on a decadal time scale, the consistent rise of air and ocean temperature will bring to a further higher glacier meltwater with the increase of both
TMF and clastic contribution.

## 6   Concluding remarks

Our 6-year time-series of marine particle fluxes and physical parameters in Kongsfjorden provided an extraordinary opportunity to investigate seasonal and multi-years changes in the inner-fjord. Our results suggest that, as the Arctic temperature rises in

a warming scenario, the flux of glacier-derived material will increase accordingly. In particular, our time-series points towards the subglacial runoff driven by air temperature being the dominant process affecting the glacier-fjord discharge of lithogenic material. Furthermore, also the watershed runoff is expected to increase the sediment delivery from land. The quantity of photosynthetic available radiation will likely increase due to reduced sea ice coverage. On the other hand, the increased turbidity will decrease the light penetration into subsurface water diminishing the primary production. Furthermore, water column stratification modulated by the inflow of warm Atlantic waters, especially in winter, will progressively hamper the exchange of nutrients from the bottom waters. This, in turn, will severely reduce the biological production and in particular the primary productivity in the fjord. Altogether, we infer that Kongsfjorden will experience a progressive increase of the glacier-derived fluxes and watershed runoff (i.e. lateral component) at the expenses of the in-situ production (i.e. vertical component).

*Data availability.* Total mass fluxes and content of major constituents of sediment traps samples are available in Table A1. The meteorological and hydrological time series are available upon request to Leonardo Langone (leonardo.langone@ismar.cnr.it).

*Sample availability.* Samples stored at CNR ISMAR Facilities in Bologna, Italy

*Author contributions.* L.L., S.M., F.G. and S.A. designed the experiments and, together with A.D'A. performed the sampling procedure and the measurements. A.V. and M.M. were involved in the implementation of the research by processing the meteorological data. A.D'A. carried out the preliminary treatment on sediment trap samples and together with A.S.-V. carried out the opal analysis. L.L., S.M. and A.D'A. processed the hydrological dataset. T.T., L.L., and A.D'A. were involved in the interpretation of the results. A.D'A. wrote the paper with input from all co-authors and produced the figures.

*Competing interests.* The authors declare that they have no conflict of interest.

*Acknowledgements.* We would like to thank the staff of Arctic station Dirigibile Italia and Kings Bay AS for logistic support. We greatly acknowledge the Captains and crew of the ship MS Teisten for their help during mooring deployment and recovery. We also thank Prof. Giuliana Panieri (Tromso) and Prof. Philippe Kerherve' (Perpignan) for fruitful discussions on an early version of the ms and two anonymous referees who greatly improved the final version. Dr. Kyle Mayers (NOCS) for helping in statistics. This project was funded by Progetto Premiale MIUR ARCA, coordinated by the Department of Earth Sciences and Technologies for the Environment of the Italian National Research Council. This is contribution number 1968 of CNR-ISMAR of Bologna.

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

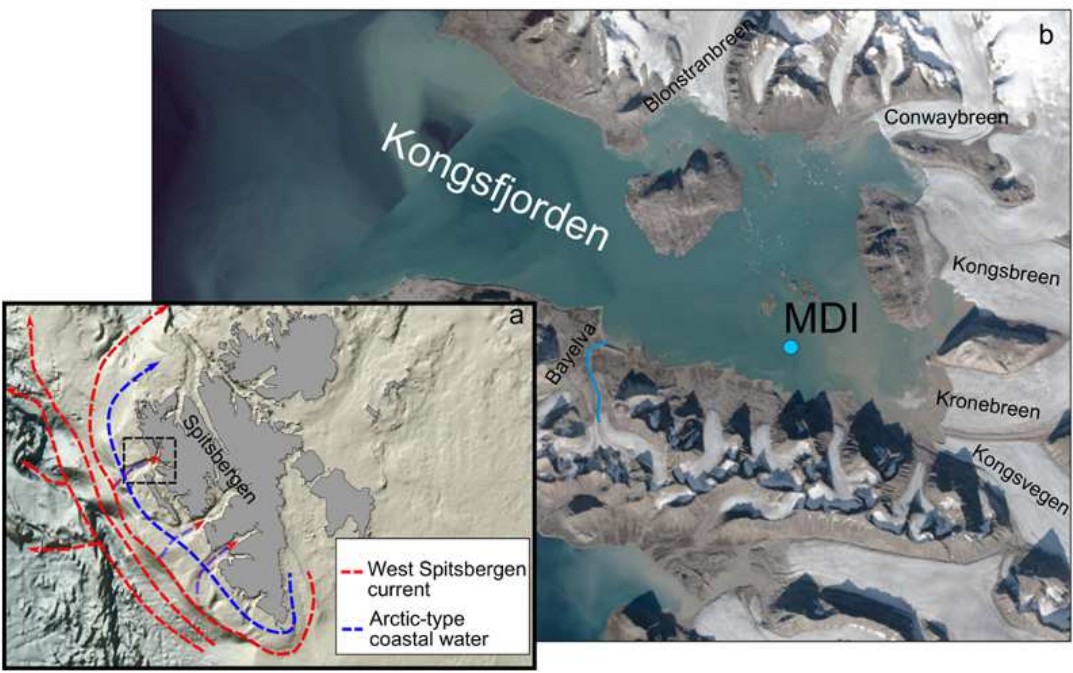

**Figure 1.** (a) Map of Svalbard with a simplified scheme of the main current circulation along the western Spitsbergen region. The light blue line shows the Arctic-type coastal water and the northward red arrow represents the warm West Spitsbergen Current (WSC). (b) Map of Kongsfjorden (area of study). The cyan point shows the Mooring Dirigibile Italia (MDI) site. MDI is located at 100 m water depth, at the distance of 1.7 km from the southern coast, between the Kronebreen front (7km far) and the sill. Position of the Bayelva river and of the main glacier fronts are also shown.

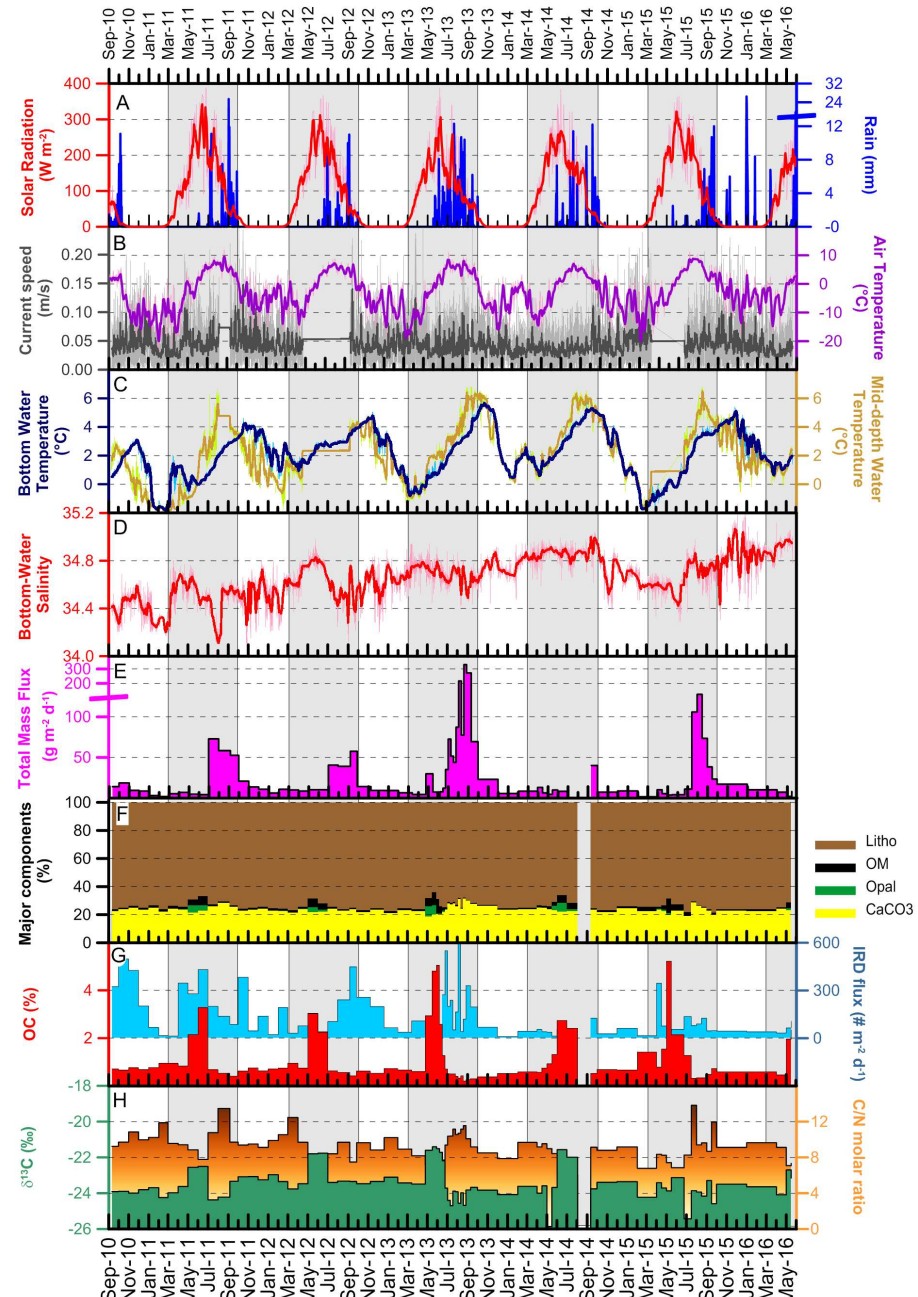

**Figure 2.** Time series of atmospheric and marine parameters recorded in Kongsfjorden between September 2010 and June 2016: (a) solar radiation and rain precipitation; (b) air temperature and mid-water current speed; (c) intermediate and near-bottom water temperature; (d) salinity at the near-bottom; (e) total mass fluxes (TMF) particle fluxes; (f) particle composition; (g) contents of OC and IRD; (h) C/N molar ratio and $\delta^{13}$C values measured on the organic fraction. Light gray bands depict the polar day of each year, characterized by enhanced solar radiation. Parameters with high-frequency temporal oscillations (solar radiation, air and water temperatures, salinity, current speed) were plotted with a superimposed week-running average (heavy lines).

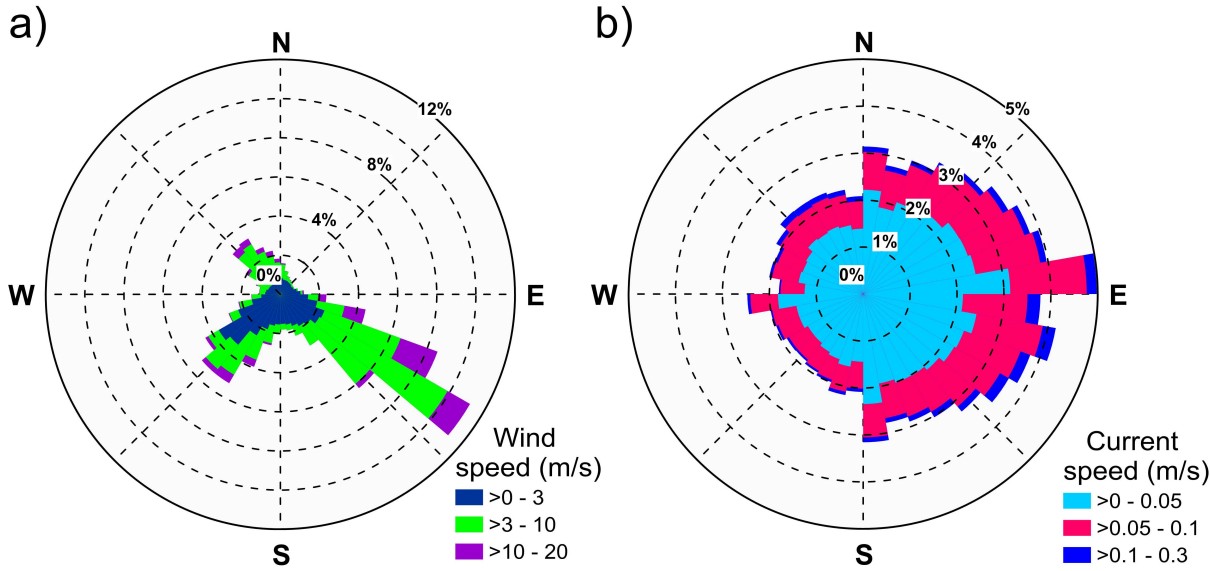

**Figure 3.** a) Inter-annual wind characteristics plotted via wind charts. b) Inter-annual ocean currents. The radius expresses the frequency of the events (%), the angle represents the geographical coordinates (degrees) and the color scale show the wind and current intensity (m s$^{-1}$).

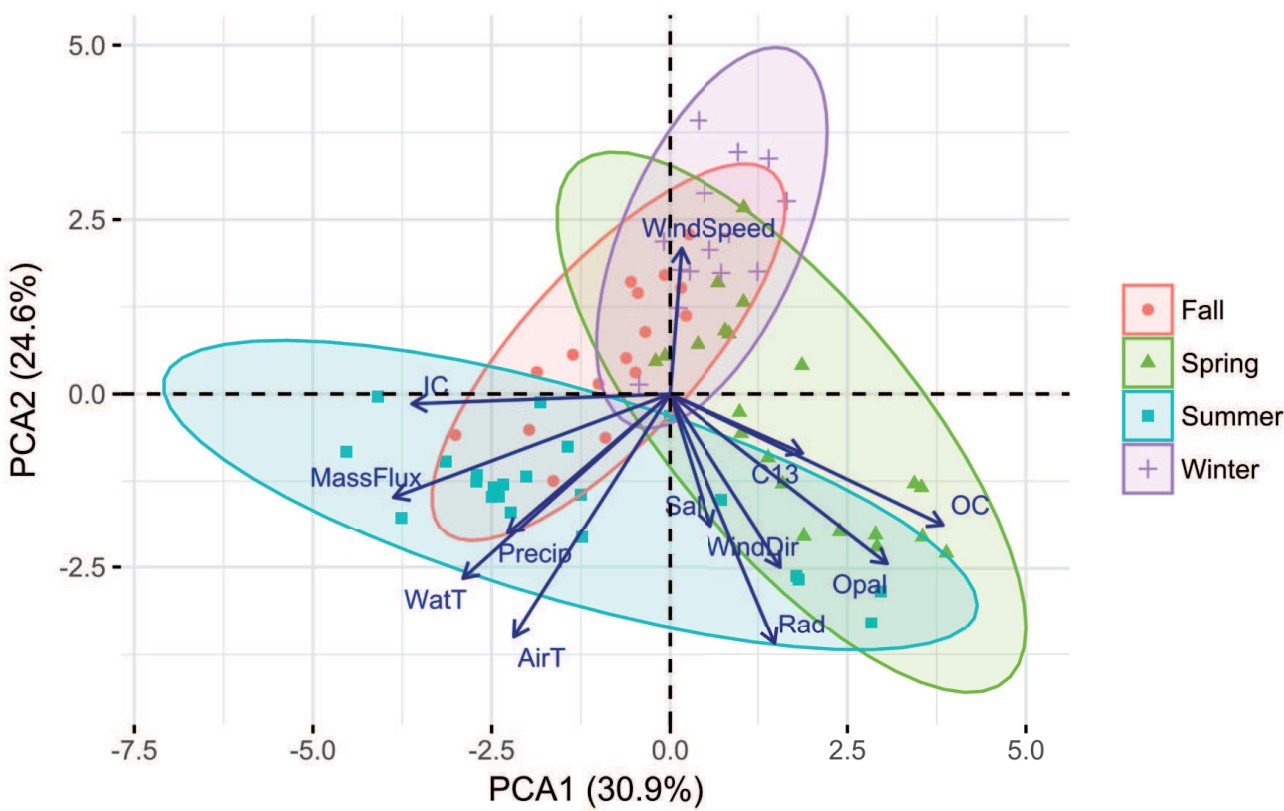

**Figure 4.** Principal Component Analysis (PCA) of the collected dataset arranged by seasons. The abbreviations are following described: total mass flux (MassFlux), organic carbon (OC), $\delta^{13}$C (C13), inorganic carbon (IC), opal (Opal), air temperature (AirT), wind speed and direction (WindSpeed, WindDir), solar radiation (Rad), rain precipitation (Precip), salinity (Sal), water temperature (WatT).

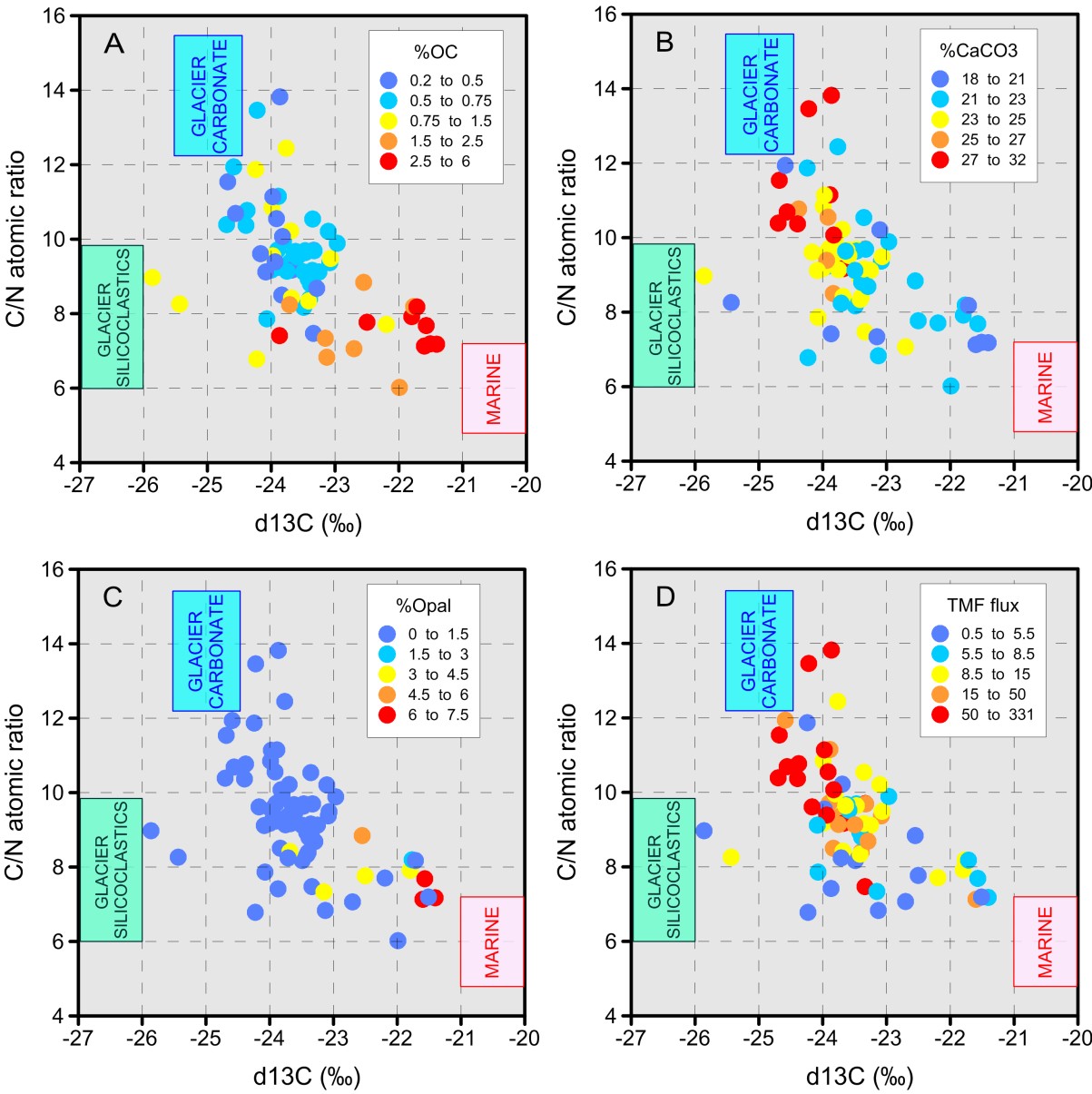

**Figure 5.** Relationship between $\delta^{13}$C and C/N for all samples plotted against %OC (a), %CaCO$_3$ (b), opal (c) and TMF (d). Coloured areas represent the estimated ranges of organic carbon end-members.

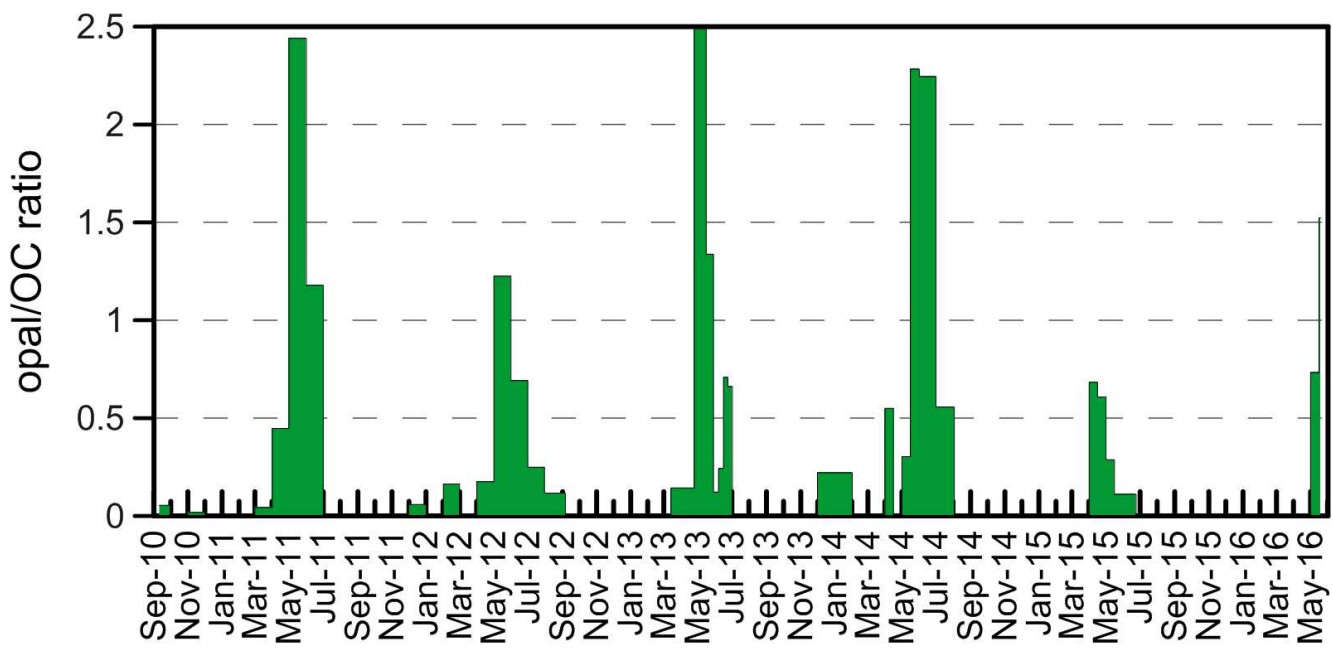

**Figure 6.** Ratio between the opal and OC contents within sediment trap samples from September 2010 to May 2016.

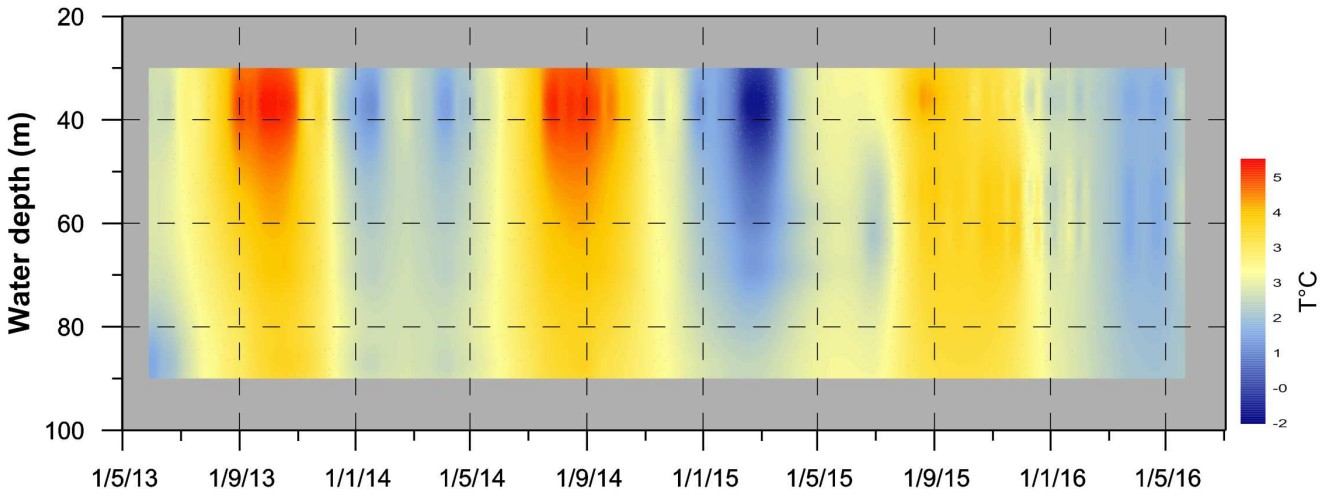

**Figure 7.** Temporal variability of the water temperature section, reconstructed between May 2013 and June 2016 based on measurements at 4 levels (36-37 m, 54-58 m, 62-69 m, 87 m).

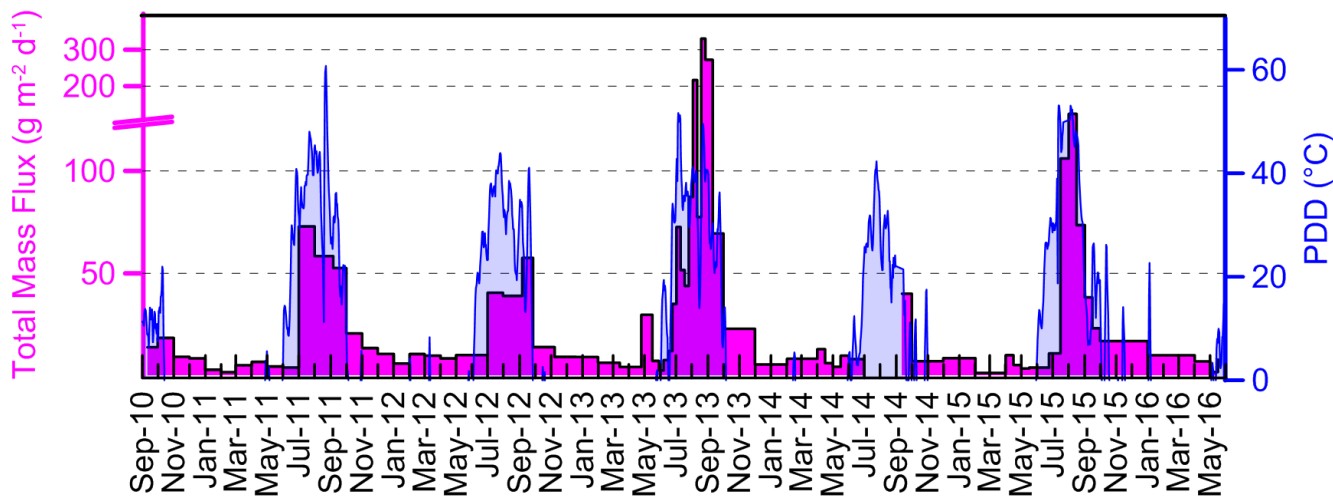

**Figure 8.** Total mass fluxes plotted against 6-day PDD sum, as a proxy of subglacial runoff.

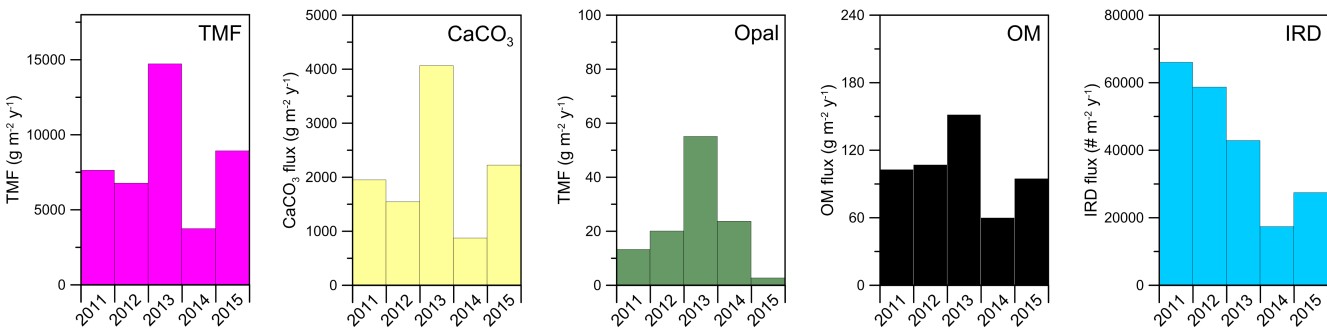

**Figure 9.** Annual fluxes of main component.

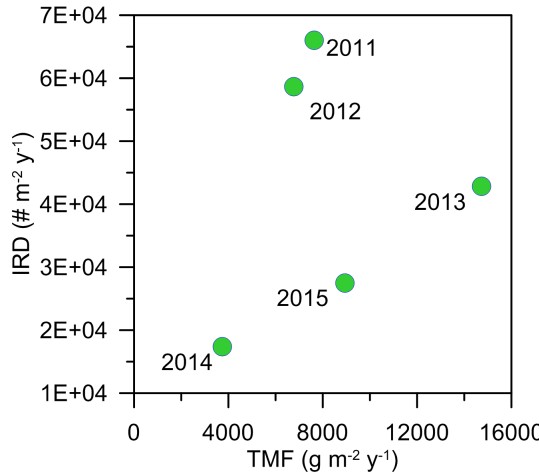

**Figure 10.** Annual TMF plotted against IRD fluxes.

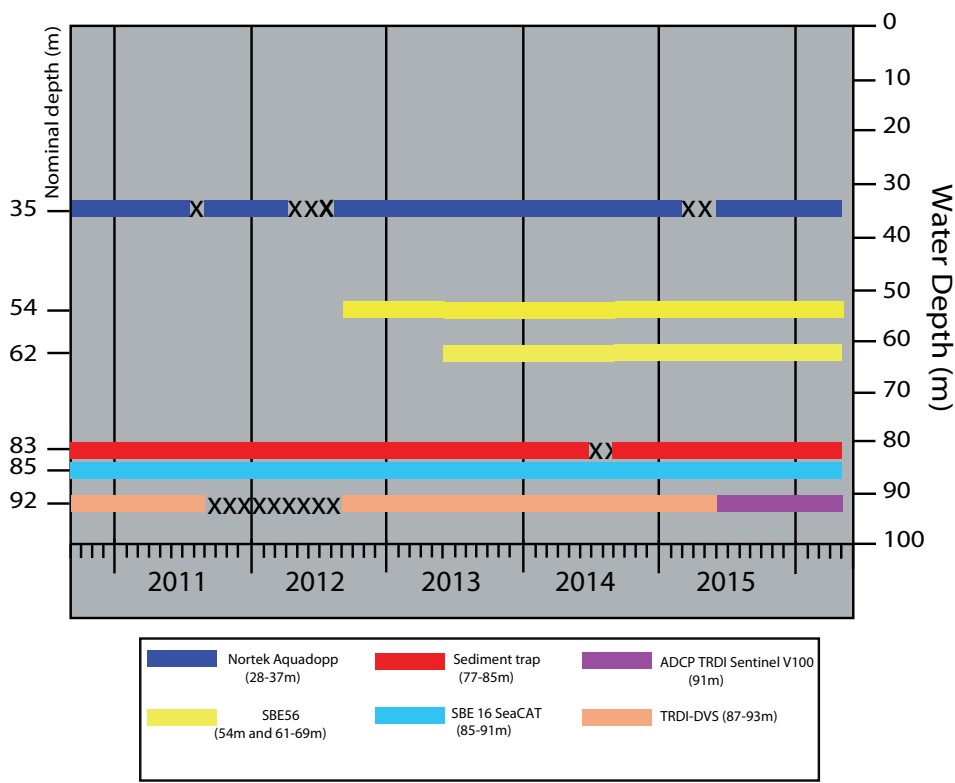

**Figure A1.** List of available time series for each instrument mounted on MDI in the period 2010-2016. No instrument was set at the surface because of the risk of damages by iceberg transit and sea ice coverage. X symbol indicates an instrument malfunctioning.

,

**Table 1.** Eigenvalues, PCA

| PC | Eigenvalues | % variation | Cumulative variation |
|----|-------------|-------------|----------------------|
| 1  | 3.71 | 30.9 | 30.9 |
| 2  | 2.95 | 24.6 | 55.5 |
| 3  | 1.16 | 9.7  | 65.2 |
| 4  | 1.02 | 8.5  | 73.7 |
| 5  | 0.79 | 6.6  | 80.3 |
| 6  | 0.69 | 5.8  | 86.1 |
| 7  | 0.6  | 5    | 91.1 |
| 8  | 0.41 | 3.4  | 94.5 |
| 9  | 0.31 | 2.6  | 97.1 |
| 10 | 0.19 | 1.6  | 98.7 |
| 11 | 0.1  | 0.8  | 99.5 |
| 12 | 0.07 | 0.5  | 100  |

**Table 2.** Coefficients in the linear combinations of variables making up PCs. The higher coefficients for each PC are shown in bold style

|  | PC1 | PC2 |
|--|-----|-----|
| OC | **0.43** | -0.24 |
| C13 | **0.21** | -0.11 |
| IC | **-0.41** | -0.02 |
| Opal | **0.34** | -0.31 |
| AirTemp | -0.25 | **-0.44** |
| Radiation | 0.17 | **-0.45** |
| Precip | -0.26 | **-0.25** |
| Salinity | 0.06 | **-0.24** |
| WaterTemp | -0.33 | **-0.34** |
| MassFlux | **-0.44** | -0.19 |
| WindSpeed | 0.02 | **0.26** |
| WindDirection | 0.18 | **-0.31** |

**Table A1.** Sediment trap dataset, with meteorological and hydrological data. Physical data were averaged according to the sampling periods.

| Sample | Start | End | MassFlux $(\text{g m}^{-2}\,\text{d}^{-1})$ | OC (%) | IC (%) | Opal (%) | airT (°C) | Radiation $(\text{W m}^{-2})$ | Precip (mm) | WindSpeed $(\text{m s}^{-1})$ | WindDir (°N) | Salinity | WatT (°C) |
|---|---|---|---|---|---|---|---|---|---|---|---|---|---|
| MDI-I-1 | 11-Sep-10 | 1-Oct-10 | 13.94 | 0.71 | 2.68 | 0.04 | 1.90 | 47.38 | 0.87 | 1.25 | 163.74 | 34.37 | 2.33 |
| MDI-I-2 | 1-Oct-10 | 1-Nov-10 | 18.49 | 0.66 | 2.83 | 0.00 | -3.27 | 6.27 | | 1.63 | 133.76 | 34.43 | 1.86 |
| MDI-I-3 | 1-Nov-10 | 1-Dec-10 | 9.30 | 0.76 | 2.96 | 0.01 | -11.01 | 0.15 | | 1.74 | 159.17 | 34.51 | 0.24 |
| MDI-I-4 | 1-Dec-10 | 1-Jan-11 | 8.53 | 0.68 | 2.80 | 0.00 | -10.66 | 0.22 | | 2.37 | 126.77 | 34.41 | -0.68 |
| MDI-I-5 | 1-Jan-11 | 1-Feb-11 | 2.96 | 0.77 | 2.97 | 0.00 | -13.63 | 0.20 | | 2.89 | 145.87 | 34.30 | -1.60 |
| MDI-I-6 | 1-Feb-11 | 1-Mar-11 | 1.78 | 0.95 | 2.64 | 0.00 | -9.71 | 2.96 | | 3.43 | 142.10 | 34.29 | -1.76 |
| MDI-I-7 | 1-Mar-11 | 1-Apr-11 | 5.19 | 0.95 | 2.89 | 0.04 | -12.23 | 46.54 | | 1.85 | 128.85 | 34.50 | -1.65 |
| MDI-I-8 | 1-Apr-11 | 1-May-11 | 6.82 | 0.84 | 2.81 | 0.37 | -4.61 | 107.82 | | 3.19 | 151.35 | 34.63 | -0.80 |
| MDI-I-9 | 1-May-11 | 1-Jun-11 | 4.56 | 2.15 | 2.53 | 5.25 | -1.63 | 212.66 | | 0.90 | 204.29 | 34.60 | -0.49 |
| MDI-I-10 | 1-Jun-11 | 1-Jul-11 | 4.06 | 3.28 | 2.72 | 3.86 | 4.07 | 264.98 | 0.10 | 0.93 | 290.19 | 34.47 | 2.29 |
| MDI-I-11 | 1-Jul-11 | 1-Aug-11 | 72.88 | 0.67 | 3.08 | 0.00 | 6.79 | 191.58 | 0.77 | 0.28 | 173.88 | 34.32 | 3.51 |
| MDI-I-12 | 1-Aug-11 | 1-Sep-11 | 58.40 | 0.53 | 3.39 | 0.00 | 6.51 | 110.65 | 1.79 | 0.96 | 130.79 | 34.42 | 5.50 |
| MDI-II-1 | 6-Sep-11 | 1-Oct-11 | 52.59 | 0.41 | 3.06 | 0.00 | 3.24 | 37.69 | 1.23 | 0.82 | 190.49 | 34.55 | 3.68 |
| MDI-II-2 | 1-Oct-11 | 1-Nov-11 | 20.70 | 0.62 | 2.72 | 0.00 | -3.44 | 10.58 | | 2.46 | 128.41 | 34.47 | 2.77 |
| MDI-II-3 | 1-Nov-11 | 1-Dec-11 | 13.55 | 0.76 | 2.79 | 0.00 | -6.30 | 0.38 | | 1.66 | 166.80 | 34.53 | 1.50 |
| MDI-II-4 | 1-Dec-11 | 1-Jan-12 | 10.70 | 0.63 | 2.90 | 0.04 | -6.60 | 0.08 | | 3.46 | 103.88 | 34.58 | 1.11 |
| MDI-II-5 | 1-Jan-12 | 1-Feb-12 | 6.00 | 0.71 | 2.74 | 0.00 | -3.21 | 0.53 | | 6.01 | 91.23 | 34.50 | 0.47 |
| MDI-II-6 | 1-Feb-12 | 1-Mar-12 | 10.65 | 0.73 | 2.69 | 0.12 | -6.57 | 2.83 | | 3.17 | 152.79 | 34.50 | 0.24 |
| MDI-II-7 | 1-Mar-12 | 1-Apr-12 | 9.87 | 0.95 | 2.53 | 0.00 | -5.67 | 47.19 | | 3.28 | 137.60 | 34.61 | 1.16 |
| MDI-II-8 | 1-Apr-12 | 1-May-12 | 8.48 | 0.74 | 2.85 | 0.13 | -9.24 | 138.40 | | 1.29 | 168.86 | 34.71 | 0.56 |
| MDI-II-9 | 1-May-12 | 1-Jun-12 | 10.12 | 3.02 | 2.58 | 3.70 | -1.85 | 233.74 | | 0.52 | 157.38 | 34.80 | n.a. |
| MDI-II-10 | 1-Jun-12 | 1-Jul-12 | 10.12 | 2.24 | 2.62 | 1.55 | 3.93 | 250.51 | 0.80 | 1.48 | 234.60 | 34.75 | n.a. |
| MDI-II-11 | 1-Jul-12 | 31-Jul-12 | 40.54 | 0.63 | 2.88 | 0.16 | 6.35 | 186.35 | 0.43 | 0.47 | 61.78 | 34.55 | n.a. |
| MDI-II-12 | 31-Jul-12 | 31-Aug-12 | 39.00 | 0.56 | 2.70 | 0.07 | 5.11 | 103.06 | 0.72 | 0.93 | 31.81 | 34.49 | n.a. |
| MDI-III-1 | 6-Sep-12 | 27-Sep-12 | 57.57 | 0.42 | 2.85 | 0.00 | 2.43 | 41.34 | 0.76 | 2.65 | 113.57 | 34.54 | 4.04 |
| MDI-III-2 | 27-Sep-12 | 8-Nov-12 | 14.01 | 0.61 | 2.62 | 0.00 | -3.00 | 12.31 | | 1.46 | 146.20 | 34.59 | 3.91 |
| MDI-III-3 | 8-Nov-12 | 20-Dec-12 | 9.22 | 0.53 | 2.79 | 0.00 | -7.16 | 0.24 | | 2.58 | 148.77 | 34.59 | 2.50 |
| MDI-III-4 | 20-Dec-12 | 31-Jan-13 | 9.17 | 0.72 | 2.51 | 0.00 | -8.70 | 0.30 | | 2.33 | 151.78 | 34.66 | 1.64 |
| MDI-III-5 | 31-Jan-13 | 14-Mar-13 | 6.40 | 0.66 | 2.76 | 0.00 | -12.24 | 11.75 | | 2.12 | 148.48 | 34.67 | -0.02 |
| MDI-III-6 | 14-Mar-13 | 25-Apr-13 | 4.32 | 0.59 | 2.67 | 0.09 | -10.01 | 105.36 | | 2.80 | 137.31 | 34.76 | -0.37 |
| MDI-III-7 | 25-Apr-13 | 16-May-13 | 29.80 | 2.93 | 2.20 | 7.31 | -3.48 | 173.38 | | 1.21 | 207.63 | 34.73 | 1.51 |
| MDI-III-8 | 16-May-13 | 26-May-13 | 7.27 | 4.81 | 2.40 | 6.42 | -0.94 | 193.32 | | 0.72 | 180.80 | 34.69 | 1.91 |
| MDI-IV-1 | 29-May-13 | 7-Jun-13 | 2.64 | 5.05 | 2.42 | 0.62 | 2.51 | 239.64 | 1.45 | 3.62 | 221.53 | 34.76 | 2.13 |
| MDI-IV-2 | 7-Jun-13 | 16-Jun-13 | 7.67 | 2.58 | 2.41 | 0.63 | 0.72 | 250.72 | 1.00 | 5.16 | 289.35 | 34.68 | 2.09 |
| MDI-IV-3 | 16-Jun-13 | 24-Jun-13 | 12.11 | 1.27 | 2.59 | 0.90 | 4.42 | 149.77 | 1.82 | 1.96 | 204.73 | 34.66 | 2.01 |
| MDI-IV-4 | 24-Jun-13 | 2-Jul-13 | 35.08 | 0.71 | 2.79 | 0.47 | 6.32 | 140.74 | 4.40 | 3.47 | 183.31 | 34.63 | 2.81 |

| Sample | Start | End | MassFlux (g m$^{-2}$ d$^{-1}$) | OC (%) | IC (%) | Opal (%) | airT (°C) | Radiation (W m$^{-2}$) | Precip (mm) | WindSpeed (m s$^{-1}$) | WindDir (°N) | Salinity | WatT (°C) |
|---|---|---|---|---|---|---|---|---|---|---|---|---|---|
| MDI-IV-5 | 2-Jul-13 | 10-Jul-13 | 72.62 | 0.51 | 3.33 | 0.00 | 7.41 | 122.75 | 1.32 | 0.69 | 217.37 | 34.62 | 3.18 |
| MDI-IV-6 | 10-Jul-13 | 18-Jul-13 | 51.62 | 0.50 | 3.25 | 0.00 | 5.85 | 239.27 | 0.38 | 2.59 | 344.08 | 34.69 | 2.92 |
| MDI-IV-7 | 18-Jul-13 | 26-Jul-13 | 43.88 | 0.51 | 3.34 | 0.00 | 5.74 | 137.23 | 3.86 | 1.08 | 153.67 | 34.74 | 3.12 |
| MDI-IV-8 | 26-Jul-13 | 3-Aug-13 | 87.27 | 0.40 | 3.10 | 0.00 | 6.68 | 124.45 | 1.49 | 1.38 | 171.22 | 34.71 | 3.67 |
| MDI-IV-9 | 3-Aug-13 | 11-Aug-13 | 216.44 | 0.33 | 3.71 | 0.00 | 5.34 | 123.09 | 3.37 | 1.11 | 210.26 | 34.67 | 3.93 |
| MDI-IV-10 | 11-Aug-13 | 19-Aug-13 | 77.45 | 0.43 | 2.88 | 0.00 | 5.45 | 78.20 | 5.84 | 2.57 | 154.43 | 34.64 | 4.35 |
| MDI-IV-11 | 19-Aug-13 | 27-Aug-13 | 330.36 | 0.21 | 3.78 | 0.00 | 7.10 | 78.41 | 1.81 | 3.45 | 124.38 | 34.66 | 5.01 |
| MDI-IV-12 | 27-Aug-13 | 4-Sep-13 | 272.28 | 0.26 | 3.58 | 0.00 | 5.20 | 68.81 | 0.23 | 0.81 | 15.75 | 34.51 | 5.86 |
| MDI-V-1 | 10-Sep-13 | 1-Oct-13 | 69.46 | 0.29 | 3.30 | 0.00 | 3.80 | 49.82 | 1.37 | 2.97 | 147.81 | 34.65 | 5.94 |
| MDI-V-2 | 1-Oct-13 | 1-Dec-13 | 22.95 | 0.37 | 3.14 | 0.00 | -7.46 | 7.67 | | 0.70 | 163.38 | 34.76 | 4.98 |
| MDI-V-3 | 1-Dec-13 | 1-Feb-14 | 5.53 | 0.52 | 2.79 | 0.12 | -6.32 | 0.46 | | 3.06 | 160.50 | 34.71 | 1.25 |
| MDI-V-4 | 1-Feb-14 | 1-Apr-14 | 8.34 | 0.59 | 2.83 | 0.00 | -5.99 | 33.06 | | 2.53 | 148.07 | 34.82 | 1.84 |
| MDI-V-5 | 1-Apr-14 | 16-Apr-14 | 13.00 | 0.57 | 2.98 | 0.32 | -11.05 | 102.90 | | 0.50 | 169.59 | 34.83 | 0.88 |
| MDI-V-6 | 16-Apr-14 | 1-May-14 | 6.17 | 0.73 | 2.97 | 0.00 | -8.82 | 171.42 | | 1.22 | 275.66 | 34.88 | 1.50 |
| MDI-V-7 | 1-May-14 | 16-May-14 | 4.42 | 0.93 | 2.83 | 0.28 | -5.23 | 213.01 | | 2.46 | 163.93 | 34.90 | 2.02 |
| MDI-V-8 | 16-May-14 | 1-Jun-14 | 9.83 | 1.32 | 2.78 | 3.00 | -1.21 | 223.30 | | 2.02 | 265.17 | 34.88 | 2.35 |
| MDI-V-9 | 1-Jun-14 | 1-Jul-14 | 8.14 | 2.74 | 2.65 | 6.14 | 1.84 | 234.71 | 0.25 | 1.42 | 243.25 | 34.87 | 3.47 |
| MDI-V-10 | 1-Jul-14 | 3-Aug-14 | 0.50 | 2.41 | 2.66 | 1.34 | 5.26 | 164.33 | 0.99 | 1.05 | 224.23 | 34.83 | 5.24 |
| MDI-VI-1 | 12-Sep-14 | 1-Oct-14 | 40.07 | 0.52 | 2.80 | 0.00 | -0.46 | 33.78 | | 3.04 | 242.34 | 34.95 | 5.21 |
| MDI-VI-2 | 1-Oct-14 | 1-Dec-14 | 7.15 | 0.69 | 2.58 | 0.00 | -4.52 | 6.64 | | 2.89 | 152.37 | 34.73 | 3.40 |
| MDI-VI-3 | 1-Dec-14 | 1-Feb-15 | 8.63 | 0.66 | 2.92 | 0.00 | -7.53 | 0.42 | | 3.03 | 152.25 | 34.70 | 1.23 |
| MDI-VI-4 | 1-Feb-15 | 1-Apr-15 | 1.42 | 1.42 | 2.69 | 0.00 | -10.02 | 25.04 | | 4.68 | 128.58 | 34.60 | -1.08 |
| MDI-VI-5 | 1-Apr-15 | 16-Apr-15 | 10.07 | 0.79 | 2.80 | 0.54 | -4.82 | 103.48 | | 2.16 | 156.52 | 34.60 | n.a. |
| MDI-VI-6 | 16-Apr-15 | 1-May-15 | 5.28 | 1.55 | 2.68 | 0.94 | -6.23 | 166.76 | | 2.12 | 172.41 | 34.57 | n.a. |
| MDI-VI-7 | 1-May-15 | 16-May-15 | 3.58 | 5.22 | 2.33 | 1.49 | -4.10 | 204.77 | | 0.60 | 220.29 | 34.60 | n.a. |
| MDI-VI-8 | 16-May-15 | 1-Jun-15 | 4.00 | 2.14 | 2.70 | 0.24 | 0.28 | 261.41 | 0.89 | 0.50 | 71.20 | 34.50 | n.a. |
| MDI-VII-1 | 24-Jun-15 | 16-Jul-15 | 10.96 | 1.27 | 2.27 | 0.00 | 6.42 | 211.80 | 0.13 | 2.91 | 101.70 | 34.74 | 3.01 |
| MDI-VII-2 | 16-Jul-15 | 1-Aug-15 | 106.05 | 0.32 | 3.43 | 0.00 | 8.70 | 242.10 | 0.00 | 1.83 | 168.01 | 34.81 | 3.65 |
| MDI-VII-3 | 1-Aug-15 | 16-Aug-15 | 123.47 | 0.33 | 3.13 | 0.00 | 7.99 | 128.66 | 0.14 | 1.30 | 168.43 | 34.73 | 5.07 |
| MDI-VII-4 | 16-Aug-15 | 1-Sep-15 | 73.54 | 0.34 | 2.98 | 0.00 | 4.67 | 77.70 | 2.44 | 2.12 | 177.84 | 34.71 | 5.72 |
| MDI-VII-5 | 1-Sep-15 | 16-Sep-15 | 38.32 | 0.47 | 2.73 | 0.00 | 2.33 | 57.50 | 2.93 | 1.26 | 202.90 | 34.74 | 4.87 |
| MDI-VII-6 | 16-Sep-15 | 1-Oct-15 | 23.24 | 0.73 | 2.38 | 0.00 | 2.24 | 30.14 | 0.00 | 3.09 | 153.43 | 34.76 | 4.26 |
| MDI-VII-7 | 1-Oct-15 | 1-Jan-16 | 16.96 | 0.59 | 2.72 | 0.00 | -3.60 | 2.31 | | 3.26 | 160.06 | 34.84 | 2.98 |
| MDI-VII-8 | 1-Jan-16 | 1-Apr-16 | 10.03 | 0.59 | 2.69 | 0.00 | -5.27 | 16.88 | | 1.88 | 191.37 | 34.84 | 1.40 |
| MDI-VII-9 | 1-Apr-16 | 1-May-16 | 7.05 | 0.46 | 2.89 | 0.00 | -5.84 | 132.94 | | 2.28 | 134.47 | 34.94 | 0.98 |
| MDI-VII-10 | 1-May-16 | 16-May-16 | 1.85 | 1.95 | 2.77 | 1.43 | 0.69 | 155.35 | 1.00 | 3.01 | 203.67 | 34.97 | 1.66 |
| MDI-VII-11 | 16-May-16 | 19-May-16 | 7.95 | 2.35 | 2.49 | 3.57 | 0.06 | 182.78 | 0.00 | 0.47 | 180.52 | 34.95 | 2.38 |