# Peer review of "Multi-year particle fluxes in Kongsfjorden, Svalbard"

_Biogeosciences, 2018_

## Referee Comment (RC1) · Anonymous Referee #1 · 18 May 2018

This is a very good manuscript on an important topic in Arctic climate change reserach - congratulations. The authors used a unique dataset from Kongsfjord and analysed it on a high level and to an impressive depth. The ms is well written even though quite long. See my recommendations on this below. I highly recommend to publish this paper because it is not only importnat with respect to biogeochemical issues but also to many other disciplines in Arctic coastal reaserch.

My specifcic answeres to the review questions.

1. Does the paper address relevant scientific questions within the scope of BG? Yes, the ms gives a deep insight in some major biogeochemical issues with respect to temporal and spatial matter fluxes in an Arctic fjord system located almost in the center of global change, in Svalbard.

2. Does the paper present novel concepts, ideas, tools, or data? Yes, the ms presents data from a 6 year study of moording data which are new and most important for a better understanding of the effects of glacier melting and changes in the overall hydrography due to increasing air and water temperatures.

3. Are substantial conclusions reached? Yes, the ms provides sound and substantial conclusions.

4. Are the scientific methods and assumptions valid and clearly outlined? Yes, all methods and assumptions are clearly outlined and explained. This is most important because the results are also of importance for other disciplines than biogeochemistry.

5. Are the results sufficient to support the interpretations and conclusions? Yes, the results are sufficient and all interpretations and conclusions are well supported by the data presented.

6. Is the description of experiments and calculations sufficiently complete and precise to allow their reproduction by fellow scientists (traceability of results)? Yes.

7. Do the authors give proper credit to related work and clearly indicate their own new/original contribution? Yes. There are some newer articles about the inner Kongsfjord system which however mainly focus on the very shallow areas, where we see very similar results. These articles may be cited but the ms is well enough also without.

8. Does the title clearly reflect the contents of the paper? Yes

9. Does the abstract provide a concise and complete summary? Yes

10. Is the overall presentation well structured and clear? Yes

11. Is the language fluent and precise? Yes

12. Are mathematical formulae, symbols, abbreviations, and units correctly defined and used? Yes

13. Should any parts of the paper (text, formulae, figures, tables) be clarified, reduced, combined, or eliminated? The introduction, material and methods and result section are quite good written, not too long and quite clear. The discussion section is a little bit long even though well written. I recommend to go over the discussion section again and try to shorten it here and there without losing substance. The same with the graphs. The ms has a total of 12 graphs. I've read the ms without the graphs at all and understood most of the parts. I therefore recommend e.g. to summarize some information's in the graphs which do not really provide absolutely important information. My favorites in this point are e.g. graph 4 and 5 which show wind and current in the different years. I think that nobody will try to extract data from this graphs from the different year so the years might be averaged to one graph because the necessary patterns among the years are provided also in the text.

14. Are the number and quality of references appropriate? Yes

15. Is the amount and quality of supplementary material appropriate? Yes

Further very specific comments on the ms are: Page 4, line 6: I do not understand 5°C as unit for direction measurement? Page 4, line 17: A dot is missing after (Karl and Knauer, 1989) Page 7, line 1: This wind conditions however have completely different effects on the surface waters because the effecvtive area where the wind is able to affect the water surface is much larger. Therefore, intense wave formation happen only during these wind conditions while even much stronger wind coming from the south have less effects on the surface waters.

---

## Referee Comment (RC2) · Anonymous Referee #2 · 22 May 2018

I think that it is a scientifically sound study, providing some important insight into sediment fluxes in a rapidly changing fjord in the Arctic. It is based on a relatively "long" time series and it helps understand the role of the glacier discharges in shaping the sediment fluxes within the fjord and also makes some "informed" speculations about expected trends in those fluxes. There are some issues that I believe may require some further clarification in the text and I hope that my comments may be of some use for that. I have made a number of small corrections and added some comments/questions to the pdf version of the manuscript that I uploaded. I think this paper is well written, with the exception of some minor things that are very easy to correct. 1. Does the paper address relevant scientific questions within the scope of BG? Yes 2. Does the paper present novel concepts, ideas, tools, or data? Especially

new data 3. Are substantial conclusions reached? Yes 4. Are the scientific methods and assumptions valid and clearly outlined? Yes 5. Are the results sufficient to support the interpretations and conclusions? Yes 6. Is the description of experiments and calculations sufficiently complete and precise to allow their reproduction by fellow scientists (traceability of results)? Not always. I made some comments about this, especially in what concerns end-members shown in Figure 7. 7. Do the authors give proper credit to related work and clearly indicate their own new/original contribution? Yes. 8. Does the title clearly reflect the contents of the paper? Yes 9. Does the abstract provide a concise and complete summary? Yes 10. Is the overall presentation well-structured and clear? Yes 11. Is the language fluent and precise? Yes, except for some minor problems 12. Are mathematical formulae, symbols, abbreviations, and units correctly defined and used? Yes 13. Should any parts of the paper (text, formulae, figures, tables) be clarified, reduced, combined, or eliminated? Small clarifications indicated in the ms. 14. Are the number and quality of references appropriate? Yes 15. Is the amount and quality of supplementary material appropriate?

Please also note the supplement to this comment:
https://www.biogeosciences-discuss.net/bg-2018-174/bg-2018-174-RC2-supplement.pdf
* * *
[Figure]

**Supplement:**

**Multi-year particle fluxes in Kongsfjorden, Svalbard**

Alessandra D'Angelo1,2, Federico Giglio1, Stefano Miserocchi1, Anna Sanchez-Vidal3, Stefano Aliani1, Tommaso Tesi1, Angelo Viola4, Mauro Mazzola4, and Leonardo Langone\*1 1CNR-ISMAR - Consiglio Nazionale delle Ricerche - Istituto di Scienze Marine, Italy 2Università degli Studi di Siena, Siena, Italy 3Universitat de Barcelona, Barcelona, Spain 4CNR-ISAC - Consiglio Nazionale delle Ricerche - Istituto di Scienze dell'Atmosfera e del Clima, Italy **Correspondence:** \*Leonardo Langone (leonardo.langone@ismar.cnr.it)

**Abstract.** High latitude regions are warming faster than other areas due to reduction of snow cover, sea ice loss, changes in atmospheric and ocean circulation. The combination of these processes, collectively known as polar amplification, provides an extraordinary opportunity to document the ongoing thermal destabilisation of the terrestrial cryosphere and the release of land-derived material into the aquatic environment. This study presents a six-year time-series (2010-2016) of physical

- 5 parameters and particles fluxes collected by an oceanographic mooring in Kongsfjorden (Spitsbergen, Svalbard). In recent decades, Kongsfjorden has been experiencing rapid loss of sea ice coverage and retreat of local glaciers as a result of the progressive increase of ocean and air temperatures. The overarching goal of this study was to continuous monitoring the inner fjord particle sinking and to understand to what extent the temporal evolution of particulate fluxes were linked to the progressive changes in both Atlantic and freshwater input. Our data show high peaks of settling particles during warm seasons,
- 10 in terms of both organic and inorganic matter. The different sources of suspended particles were described as a mixing of glacier carbonate, glacier-silicoclastic and autochthonous marine input. The glacier releasing sediments into the fjord resulted to be the predominant source, while the sediment input by rivers was reduced at the mooring site. Our time-series showed that the seasonal sunlight exerted first-order control on the particulate fluxes in the inner fjord. The marine fraction peaked when the solar radiation was maxima in May-June while the land-derived fluxes exhibited a 1-2 months lag consistent with the maximum
- 15 air temperature and glacier melting. The inter-annual time-weighted total mass fluxes varied two-order of magnitudes over time, with relatively higher values in 2011, 2013 and 2015. Our results suggest that the land-derived input will remarkably increase over time in a warming scenario. Further studies are therefore needed to understand the future response of the Kongsfjorden ecosystem alterations in respect to the enhanced release of glacier-derived material.

20 There is ample evidence collected over the last decades that the atmosphere and ocean have warmed, the amounts of snow and ice have diminished and sea level has risen (IPCC, 2014). Global climate change is amplified in the Arctic by several positive feedbacks, including ice and snow melting that decreases surface albedo and atmospheric stability that traps temperature anomalies near the surface layers (Overpeck et al., 1997).

The physical drivers for the changes attributed to anthropogenic climate change include the increased penetration of warm Atlantic and Pacific water into the Arctic Ocean, increased seawater temperature, reduced cover of sea ice and increased submarine irradiance (Wassmann et al., 2011). As a result, the temperature in high latitudes is increasing at a rate of two to three times than the global average temperature (ACIA, 2004). Arctic fjord systems will likely be particularly vulnerable to the human induced climate change

5 human-induced climate change.

To better understand how the thermal destabilization of Arctic fjords will occur over time, it is important to establish the current knowledge for these sites. This requires the acquisition of time series for climate-sensitive parameters (Svendsen et al., 2002). Our research is part of the ARCA project (ARctic present Climate change and pAst extreme events), which aimed to develop a conceptual model on the mechanism(s) behind the release of large volumes of cold and fresh water from melting of

- 10 ice caps, investigating this complex system from both paleoclimatic and modern air-sea-ice interaction process point of view.
- Six-years continuous data (2010-2016) have been collected using an automatic sediment trap moored in the inner Kongsfjorden. The mooring line was also equipped with current meters, salinity and temperature sensors. The mooring location was chosen based on a detailed morpho-bathymetric survey, together with hydrologic investigations. It is located between the glaciers termini and the sill receiving the influence by melt water from the glacier as well as the Atlantic Water (AW) intrusion
- **15** through the southern fjord (Svendsen et al., 2002; Cottier et al., 2005).

In this study geochemical data of sinking particles were combined with the time-series of physical environmental data measured in the inner fjord. In order to describe the downward particle fluxes of biogenic and glacier-derived material, a geochemical toolbox provided of bulk organic matter geochemistry (organic carbon, total nitrogen and  $\delta^{13}$ C), inorganic carbon and opal contents, as well as Ice Rafted Detritus (IRD) concentration was used.

[revised manuscript text omitted]
 at tidewater glaciers. The frontal ablation is not dependent on ice dynamics, nor reduced by glacier surface freeze-up, but varies strongly with sub-surface water temperature. In Kongsfjorden, submarine melting is specifically forced by the AW intrusions. However, the delay of maximum water temperature (Fig. 3c) from the TMF peaks (Fig. 3f) would suggest that submarine melting is a minor font of sediment for site MDI, and can at best explain the slow declining trend of TMF during some falls (e.g., in 2011,

15 2013 and 2015; Fig. 3f).

Marine currents measured at mid-depth of mooring MDI were mostly directed toward the inner fjord (Fig. 5), whereas winds coming from the glacier valleys (Fig. 4) force the surface water to move out of the fjord (Aliani et al., 2016). It results that water circulation in the innermost part of Kongsfjorden is of estuarine-like type with cold freshwater on top of warm saline water (Svendsen et al., 2002; Cottier et al., 2005; Trusel et al., 2010; Aliani et al., 2016). A high resolution numerical

- 20 ocean circulation model has recently further showed a strong surface outflow of glacier runoff in summer compensated by a sub-surface inflow near the glacier fronts (Sundfjord et al., 2017). This type of circulation implies that the glacially-derived sediment will spread at surface through hypopycnal sediment-rich plumes dispersed by winds over the fjord. Specifically, the primary driver of hypopycnal plume formation is a meltwater upwelling from Kronebreen (Trusel et al., 2010). As meltwater discharges from the base of the glacier, it entrains high concentrations of sediment and rapidly forms a turbulent jet (Powell,
- 1991). Because of relative density contrasts, the jet rises vertically and forms a buoyant and brackish surface overflow plume.At the surface, the brackish plume spreads then laterally (Trusel et al., 2010) driven by the wind.

Finally, PCA results (Fig. 6) suggest that even the surface runoff by rainfalls can increase the sediment delivery to the fjord, especially in summer 2013 (Fig. 3a), making the upper water very turbid and resulting in typical red coloured waters. Winter precipitation events in Ny-Ålesund have been increasing during recent years (on avg. 3-4% per decade for the 1961 - 2010

30 period (Førland et al., 2011). As a matter of facts, the concurrent air temperature increase may enhance the rain fraction against snow. The rain events recorded in Jan-Feb 2016 (Fig. 3a) testify the incursion of mild and wetter air masses into the fjord. Therefore, in a warming scenario, an increase of particle flux to Kongsfjorden by local watersheds can be expected, even during winter.

**5.5 Annual fluxes and possible changes**

TMFs measured at the MDI station are about 20-100 times higher compared to downward fluxes measured on the Spitsbergen continental margin (Sanchez-Vidal et al., 2015; Lalande et al., 2016b), but of the same order of those measured in August 2012 in Kongsfjorden by (Lalande et al., 2016a). For studying the variability of particle fluxes at multi-year scale the use of annual

5 time-averaged TMFs is preferable. The lowest time-weighted averages were calculated for 2010, 2014 and 2016  $(4.6 \times 10^3; 3.8 \times 10^3; 3.1 \times 10^3 \text{ g m}^{-2} \text{ y}^{-1}$ , respectively), but these time-weighted TMFs could be underestimated because the time-series were not complete and missed the summer, the most abundant flux season. The inter-annual time-weighted TMFs varied of a factor 2 over time ( $6.8 \times 10^3$  to  $14.7 \times 10^3$  g m-2 for 2012 and 2013, respectively).

Annual total mass and main component fluxes are rather constant over time (Fig. 11), with the exception of 2013, when

[revised manuscript text omitted]

---

## Author Comment (AC1) · 3 Jul 2018

I'd like to thank you for the helpful comments. Following, the answers for each point.

Answers to comments at "Discussion" section. The discussion section is a little bit long even though well written. I recommend to go over the discussion section again and try to shorten it here and there without losing substance.

The discussion section was slightly reduced in length, trying to maintain the original clarity.

Answers to comments at figures. To reduce the number of graphs. The ms has a total of 12 graphs.

Figures were reduced in number. Now they are 10 (2 less than the first version): the original "Fig. 2" is now "Fig. A1", available in the Annexes section; Figs. 4 and 5 are now Fig. 3a and b. Here, it is shown the averaged time-series wind and current direction data in two graphs, respectively. Consequently, the numbering of all the figures was changed.

Minor comments: Page 4, line 6: I do not understand 5°C as unit for direction measurement?

It was just a typo error, we corrected it.

Page 4, line 17: A dot is missing after (Karl and Knauer, 1989).

Corrected.

Page 7, line 1: This wind conditions however have completely different effects on the surface waters because the effective area where the wind is able to affect the water surface is much larger. Therefore, intense wave formation happen only during these wind conditions while even much stronger wind coming from the south have less effects on the surface waters.

We agree with it.

Please also note the supplement to this comment:
https://www.biogeosciences-discuss.net/bg-2018-174/bg-2018-174-AC1-supplement.pdf

---

## Author Comment (AC2) · 3 Jul 2018

We'd like to thank you for the helpful comments. Following, the answers for each point.

Answers to comments at "Introduction" section.

The Introduction seems rather small lacking references to similar works done elsewhere in the Arctic.

We agree with this comment, thus we added some references describing the influence on particles settling of tidal glaciers surrounding the fjord.

I think this is too detailed for the Introduction for it belongs to the Methods section, especially the second underlined paragraph.

[Figure]

We reduced the methodological description of the mooring and left what strictly need to explain the aims of the investigation.

Also, there seems to be something wrong with the sentence "geochemical toolbox provided of...". Please check.

We removed the sentence.

I guess that six years is certainly a long time-series but perhaps not enough to say much about global change patterns.

We smoothed the assertion.

Answers to comments at "Materials and methods" section.

- 3.2 Trap sample treatment and analytical methods: the sentence "The total weight of the trapped sediment was converted to flux according to each sample duration and to the trap collection area" was moved above, as suggested by the reviewer.

- 3.3 Principal component analysis (PCA): a sentence was added to explain how the dataset was transformed (Standardized to a mean of 0 and standard deviation of 1).

Answers to comments at "Results" section.

- 4.5 Oceanography: I can see the negative values in 2012 but not in 2014. May you confirm please?

We used "negative peak" to mean two minimum values, not necessary temperature values below zero. This caused misunderstanding. To better clarify, we modified the word "negative" in "cold".

- 4.8 PCA: One can see the loadings in the Table 2 and wind speed is opposite to all the other. However, from this sentence it is unclear which opposes to which. I suggest rephrasing.

We rephrased the sentence as follows: "The main coefficient for PC2 is the wind speed,

whereas air and water temperatures, radiation, precipitation, (weakly) salinity and wind direction are in opposite loading."

Answers to comments at "Discussion" section.

- 5.1 Seasonal variability of particle fluxes: I suppose that wind direction may be a bit misleading in this analysis because extremely different values may reflect very similar directions when the wind comes from the north.

Honestly, we did not understand the comment, but it does not seem critical for the overall comprehension of the text.

- 5.2 Nature of collected particles: Why intriguingly? I can see that Burgeois et al. (2016) found higher delta13C values in sediments of their inner station. Also, Kumar et al. (2016) found basically the same along the fjord axis, with delta13C values increasing towards the inner station. Both authors found also higher C:N ratios in the inner fjord. So, I guess there is some consistency regarding the sediments: those near the glacier front are apparently more enriched in 13C and impoverished in particulate nitrogen than those closer to the ocean. I can also see that values presented by Calleja et al. (2017) for suspended matter do not show any clear trend from the inner to the outer fjord regarding the delta13C but they show that C:N ratios are lowest for the inner station which matches the results of the previous authors. So, the fact that you have a negative relationship between C:N and delta13C (Figure 7) seems to contradict the results presented by the previous authors. I wonder if this results from the fact that all your measurements come from the same point in the fjord and because of that all your variability is "temporal" as you discuss a bit further down.

The term "intriguingly" was used as in the Kongsfjorden, Burgeois et al. (2016) and Kumar et al. (2016) found uncommon light delta13C values for marine organic carbon and uncommon heavy delta13C values for terrestrial organic carbon. This is in contrast with almost all previous studies in the world. Actually, different authors found different ranges of delta13C values in Kongsfjorden, which makes this parameter very problematic to use in a mixing model to infer the origin of organic matter. Furthermore, many different sources of OM (marine, terrestrial by glaciers, surface runoff of permafrost, coal, coastal macroalgae, kerogene, etc.) were suggested in previous works, each one characterized by different delta13C or C/N values. Hence, we decided to just describe the overall temporal trend, minimizing the interpretation of its origin. Anyway, we deleted the term "intriguingly".

You should explain how the ranges of values defining the various end-members were defined.

The areas for each end end-member were defined based on reference values and by the distribution pattern of our data (quasi-triangular dispersion). We are aware this is a rather simplified approach. However, we did not use the end-member composition for any mixing model, our aim was just to make general inferences on the nature of collected particles.

The values I see in Figure 7 for OC associated with these "glacier" end-members seem rather low...why do you say values are relatively high? Also, in the PCA OC is opposed to variables associated with the glacier discharges (along PC1). The referee made this comment to the sentence: "The third end-member remains, though, elusive. Low opal contents do not support the hypothesis of in-situ diatom production while the relatively high OC content would suggest glacier outïnĆows quantitatively enriched in fossil/subfossil bioavailable carbon...". This was exclusively referred to the third end-member (silicoclastic rich), which has 3 typical samples (the yellow ones in Fig. 5 – new version) with a relatively high OC content (in a range of 0.75-1.5%).

- 5.5 Annual fluxes and possible changes: "Annual total mass and main component fluxes are rather constant over time (Fig. 11), with the exception of 2013". The referee comment was: "This is not what Figure 11 suggests, with major inter-annual differences. 2015 seems also quite different...".

We agree with this comment, as the sentence can be misunderstood. Thus, we

rephrased it.

Please also note the supplement to this comment:
https://www.biogeosciences-discuss.net/bg-2018-174/bg-2018-174-AC2-supplement.pdf

---

## Author Comment (AC3) · 3 Jul 2018

[revised manuscript text omitted]

,

**Table 1.** Eigenvalues, PCA

[revised manuscript text omitted]

---

## Author Response (AR2)

**AC to RCs 1**

I'd like to thank you for the helpful comments. Following, the answers for each point.

**Answers to comments at "Discussion" section.**

1) *The discussion section is a little bit long even though well written. I recommend to go over the discussion section again and try to shorten it here and there without losing substance.*
2) The discussion section was slightly reduced in length, trying to maintain the original clarity.

**Answers to comments at figures.**

1) *To reduce the number of graphs. The ms has a total of 12 graphs.*
2) Figures were reduced in number.
3) Now they are 10 (2 less than the first version): the original "Fig. 2" is now "Fig. A1", available in the Annexes section; Figs. 4 and 5 are now Fig. 3a and b. Here, it is shown the averaged time-series wind and current direction data in two graphs, respectively. Consequently, the numbering of all the figures was changed.

**Minor comments:**

1) *Page 4, line 6: I do not understand 5∘C as unit for direction measurement?*
2) It was just a typo error, we corrected it.
1) *Page 4, line 17: A dot is missing after (Karl and Knauer, 1989).*
2) Corrected.
1) *Page 7, line 1: This wind conditions however have completely different effects on the surface waters because the effective area where the wind is able to affect the water surface is much larger. Therefore, intense wave formation happen only during these wind conditions while even much stronger wind coming from the south have less effects on the surface waters.*
2) We agree with it.

**AC to RCs 2**

We'd like to thank you for the helpful comments. Following, the answers for each point.

**Answers to comments at "Introduction" section.**

1) *The Introduction seems rather small lacking references to similar works done elsewhere in the Arctic.*
2) We agree with this comment.
3) Thus, we added some references describing the influence on particles settling of tidal glaciers surrounding the fjord.
1) *I think this is too detailed for the Introduction for it belongs to the Methods section, especially the second underlined paragraph.*
2) We reduced the methodological description of the mooring and left what strictly need to explain the aims of the investigation.
1) *Also, there seems to be something wrong with the sentence "geochemical toolbox provided of...". Please check.*
2) We removed the sentence.
1) *I guess that six years is certainly a long time-series but perhaps not enough to say much about global change patterns.*
2) We smoothed the assertion.

**Answers to comments at "Materials and methods" section.**

**3.2 Trap sample treatment and analytical methods:**

1) *As I understand this sentence and equation should be before the description of the procedure applied to the remaining aliquots for clarity.*
2) the sentence *The total weight of the trapped sediment was converted to flux according to each sample duration and to the trap collection area* was moved above, as suggested by the reviewer.

**3.3 Principal component analysis (PCA):**

1) *Transformed how? Standardized by the mean and the standard deviation?*
2) A sentence was added to explain how the dataset was transformed.
3) *Standardized to a mean of 0 and standard deviation of 1.*

**Answers to comments at "Results" section.**

**4.5 Oceanography:**

1) *I can see the negative values in 2012 but not in 2014. May you confirm please?*
2) We used "negative peak" to mean two minimum values, not necessary temperature values below zero. This caused misunderstanding.
3) To better clarify, we modified the word "negative" in "cold".

**4.8 PCA**:

1) *One can see the loadings in the Table 2 and wind speed is opposite to all the other. However, from this sentence it is unclear which opposes to which. I suggest rephrasing.*
2) We rephrased the sentence as follows.
3) The main coefficient for PC2 is the wind speed, whereas air and water temperatures, radiation, precipitation, (weakly) salinity and wind direction are in opposite loading.

**Answers to comments at "Discussion" section.**

**5.1 Seasonal variability of particle fluxes:**

1) *I suppose that wind direction may be a bit misleading in this analysis because extremely different values may reflect very similar directions when the wind comes from the north.*
2) Honestly, we did not understand the comment, but it does not seem critical for the overall comprehension of the text.

**5.2 Nature of collected particles**:

1) *Why intriguingly? I can see that Burgeois et al. (2016) found higher delta13C values in sediments of their inner station. Also, Kumar et al. (2016) found basically the same along the fjord axis, with delta13C values increasing towards the inner station. Both authors found also higher C:N ratios in the inner fjord. So, I guess there is some consistency regarding the sediments: those near the glacier front are apparently more enriched in 13C and impoverished in particulate nitrogen than those closer to the ocean. I can also see that values presented by Calleja et al. (2017) for suspended matter do not show any clear trend from the inner to the outer fjord regarding the delta13C but they show that C:N ratios are lowest for the inner station which matches the results of the previous authors. So, the fact that you have a negative relationship between C:N and delta13C (Figure 7) seems to contradict the results presented by the previous authors. I wonder if this results from the fact that all your measurements come from the same point in the fjord and because of that all your variability is "temporal" as you discuss a bit further down.*
2) The term "intriguingly" was used as in the Kongsfjorden, Burgeois et al. (2016) and Kumar et al. (2016) found uncommon light $\delta^{13}$C values for marine organic carbon and uncommon heavy $\delta^{13}$C values for terrestrial organic carbon. This is in contrast with almost all previous studies in the world. Actually, different authors found different ranges of $\delta^{13}$C values in Kongsfjorden, which makes this parameter very problematic to use in a mixing model to infer the origin of organic matter. Furthermore, many different sources of OM (marine, terrestrial by glaciers, surface runoff of permafrost, coal, coastal macroalgae, kerogene, etc.) were suggested in previous works, each one characterized by different $\delta^{13}$C or C/N values. Hence, we decided to just describe the overall temporal trend, minimizing the interpretation of its origin.
3) Anyway, we deleted the term "intriguingly".

1) *You should explain how the ranges of values defining the various end-members were defined.*
2) The areas for each end end-member were defined based on reference values and by the distribution pattern of our data (quasi-triangular dispersion). We are aware this is a rather

simplified approach. However, we did not use the end-member composition for any mixing model, our aim was just to make general inferences on the nature of collected particles.

1) *The values I see in Figure 7 for OC associated with these "glacier" end-members seem rather low...why do you say values are relatively high? Also, in the PCA OC is opposed to variables associated with the glacier discharges (along PC1).* The referee made this comment to this sentence: *"The third end-member remains, though, elusive. Low opal contents do not support the hypothesis of in-situ diatom production while the relatively high OC content would suggest glacier outflows quantitatively enriched in fossil/subfossil bioavailable carbon…".*

2) This was exclusively referred to the **third end-member (silicoclastic rich),** which has 3 typical samples (the yellow ones in Fig. 5 – new version) with a relatively high OC content (in a range of 0.75-1.5%).

**5.5 Annual fluxes and possible changes:**

1) **"***Annual total mass and main component fluxes are rather constant over time (Fig. 11), with the exception of 2013*". The referee comment was: *"This is not what Figure 11 suggests, with major inter-annual differences. 2015 seems also quite different...".*

2) We agree with this comment, as the sentence can be misunderstood. Thus, we rephrased it.

**ACs to Associate Editor:**

1) *"7 km from the southern coast"*
2) We corrected the sentence.
1) *"Use the European style for dates, e.g. 3 August 2014 (or even better 2014-08-03)"*
2) We made it as suggested.
1) *"In particular, our time-series points towards the subglacial runoff driven by air temperature being the dominant process affecting the glacier-fjord discharge of lithogenic material."*
2) We rephrased it.
3) *being the dominant process…*
1) *"I could not find your data. Importantly, please provide the direct link to the data, not a link to the data portal. There should be a doi for that".*
2) Data of total mass fluxes and contents of the major constituents of sediment trap samples are fully available in Table A1. The time-series data of meteorological and hydrological parameters are freely available upon request to authors (Langone, L., leonardo.langone@ismar.cnr.it), as we are currently working to make the whole dataset available at the "Italian Arctic Data Center" maintained by CNR.
3) *Total mass fluxes and content of major constituents of sediment traps samples are available in Table A1. The meteorological and hydrological datasets time series are available upon request to Leonardo Langone (leonardo.langone@ismar.cnr.it).*

**Multi-year particle fluxes in Kongsfjorden, Svalbard**

[revised manuscript text omitted]

Six-years continuous data (2010-2016) have been collected using an automatic sediment trap moored in the inner Kongsfjorden. The mooring line was also equipped with current meters, salinity and temperature sensors. The mooring location was chosen based on a detailed morpho-bathymetric survey, together with hydrologic investigations. It is located between the glaciers termini and the sill receiving the influence by melt water from the glacier as well as the Atlantic Water (AW) intrusion through the southern fjord (Svendsen et al., 2002; Cottier et al., 2005).

In this study geochemical data of sinking particles were combined with the time series of physical environmental data measured in the inner fjord. In order to describe the downward particle fluxes of biogenic and glacier-derived material geochemical toolbox provided of bulk organic matter geochemistry (organic carbon, total nitrogen and $\delta^{13}$C), inorganic carbon and opal contents, as well as Ice Rafted Detritus (IRD) concentration was used.

[revised manuscript text omitted]